# The essential role of YAP O-GlcNAcylation in high-glucose-stimulated liver tumorigenesis

Xiao Zhang[1,*], Yongxia Qiao[2,*], Qi Wu[1], Yan Chen[1], Shaowu Zou[3], Xiangfan Liu[4], Guoqing Zhu[1], Yinghui Zhao[1], Yuxin Chen[1], Yongchun Yu[5], Qiuhui Pan[6,†], Jiayi Wang[1] & Fenyong Sun[1]

O-GlcNAcylation has been implicated in the tumorigenesis of various tissue origins, but its function in liver tumorigenesis is not clear. Here, we demonstrate that O-GlcNAcylation can enhance the expression, stability and function of Yes-associated protein (YAP), the downstream transcriptional regulator of the Hippo pathway and a potent oncogenic factor in liver cancer. O-GlcNAcylation induces transformative phenotypes of liver cancer cells in a YAP-dependent manner. An O-GlcNAc site of YAP was identified at Thr241, and mutating this site decreased the O-GlcNAcylation, stability, and pro-tumorigenic capacities of YAP, while increasing YAP phosphorylation. Importantly, we found via *in vitro* cell-based and *in vivo* mouse model experiments that O-GlcNAcylation of YAP was required for high-glucose-induced liver tumorigenesis. Interestingly, a positive feedback between YAP and global cellular O-GlcNAcylation is also uncovered. We conclude that YAP O-GlcNAcylation is a potential therapeutic intervention point for treating liver cancer associated with high blood glucose levels and possibly diabetes.

[1] Department of Clinical Laboratory Medicine, Shanghai Tenth People's Hospital of Tongji University, Shanghai 200072, China. [2] School of Public Health, Shanghai Jiaotong University School of Medicine, Shanghai 200025, China. [3] Department of Hepatopancreatobiliary, Shanghai Tenth People's Hospital of Tongji University, Shanghai 200072, China. [4] Faculty of Medical Laboratory Science, Shanghai Jiaotong University School of Medicine, Shanghai 200025, China. [5] Shanghai Municipal Hospital of Traditional Chinese Medicine, Shanghai University of Traditional Chinese Medicine, Shanghai 200071, China. [6] Department of Central Laboratory, Shanghai Tenth People's Hospital of Tongji University, Shanghai 200072, China. * These authors contributed equally to this work. † Present address: Department of Laboratory Medicine, Shanghai Children's Medical Center, Shanghai Jiaotong University School of Medicine, Shanghai 200127, China. Correspondence and requests for materials should be addressed to J.W. (email: karajan2@163.com) or to F.S. (email: sunfenyong@126.com).

O-GlcNAcylation is a specific type of posttranslational modification catalysed by O-GlcNAc transferase (OGT)[1], leading to the transfer of O-linked β-N-acetylglucosamine (O-GlcNAc) to the hydroxyl group of serine (Ser) or threonine (Thr) residues of target proteins. As both O-GlcNAcylation and phosphorylation modify Ser and/or Thr side chains of substrate proteins, they may compete for the same Ser/Thr site(s) of a substrate or, alternatively, target different Ser/Thr residues. As a consequence, O-GlcNAcylation and phosphorylation can coordinately regulate the biological function of their substrate proteins[2]. Pathologically, aberrant O-GlcNAcylation has been shown to stimulate tumorigenesis in various cancers via regulating cell signalling, transcription, cell division, metabolism and cytoskeletal regulation[3–6]. However, whether and how O-GlcNAcylation impacts liver tumorigenesis remains unclear.

The transcriptional cofactor yes-associated protein (YAP) can function to enhance transcriptional activities of multiple transcription factors, including cyclic AMP (cAMP)-response element binding protein (CREB) and TEA domain transcription factor (TEAD) families[7,8]. Its function is controlled by the tumour-suppressing Hippo pathway. The Hippo pathway consist of phosphorylation kinases macrophage stimulating (MST) 1/2 and large tumour suppressor kinase (LATS) 1/2, and their adaptor proteins, Salvador family WW domain containing protein (SAV) 1 and MOB kinase activator 1A, to form a phosphorylation kinase cascade[9–12]. The Hippo pathway kinase cascade causes phosphorylation of YAP, resulting in cytoplasmic localization of YAP and subsequent ubiquitin-mediated degradation driven by the E3-ligase ßTrCP (refs 13,14). YAP can stimulate cell proliferation and transformation[15]. In liver cancer, YAP overexpression is associated with weak tumour differentiation[16]. Transgenic overexpression of YAP leads to the dysregulation of organ size and eventual hepatocellular carcinoma in mice[17].

Diabetes-associated metabolic disorders have been established as one of the major risk factors in the progression of liver cancer[18,19]. Epidemiological studies have revealed a strong association between diabetes and the occurrence of liver cancer[20,21]. In diabetes, hyperglycaemia induces excess activity of the hexosamine biosynthesis pathway (HBP), leading to increased synthesis of UDP-N-acetyl-ᴅ-glucosamine (UDP-GlcNAc), the substrate that OGT uses for O-GlcNAcylation of target proteins[22]. This might be the molecular mechanism underlying elevated protein O-GlcNAcylation in diabetes[23,24]. Because O-GlcNAcylation plays an important role in both diabetes and cancer, O-GlcNAcylation of certain proteins may be a pivotal contributor to tumorigenesis. Moreover, several studies have also reported the relationship between diabetes and YAP[25,26]. Because of the marked increase in the prevalence of both liver cancer and diabetes[27–29], it is important to understand the molecular basis governing the functional interplay of these two diseases, as well as the potential role of YAP and O-GlcNAcylation in these diseases.

In this study, we demonstrate that YAP can be O-GlcNAcylated at Thr241. This O-GlcNAcylation event antagonizes Hippo pathway-mediated phosphorylation of YAP, thus allowing YAP to promote liver tumorigenesis under diabetes-prone, high-glucose conditions. This work also reveals the role of YAP in glucose metabolism and O-GlcNAcylation of other proteins, as well as reciprocal regulation between O-GlcNAcylation and YAP. Altogether, this study provides novel molecular insights into the initiation and progression of high-glucose-associated liver cancer.

## Results

**O-GlcNAcylation stimulates YAP activity.** By testing >200 liver cancer samples using a tissue microarray analysis (TMA), a statistically significant positive correlation between YAP expression and global O-GlcNAcylation was observed (Fig. 1a–c). TEAD is one of the most important YAP-dependent transcription factors, and its transcriptional activity relies on YAP[8,30]. Excluding the possibility that TEAD was also affected by O-GlcNAcylation is necessary to verify the conclusion that the effects of O-GlcNAcylation on liver tumorigenesis primarily occur via YAP and not via YAP-dependent transcription factors. We found that the levels of TEAD were not significantly correlated with the levels of global O-GlcNAcylation (Fig. 1a and Supplementary Fig. 1a). CTGF is a well-established YAP/TEAD-controlled gene[8], and its expression can indirectly reflect the transcriptional activity of YAP/TEAD. Therefore, we also evaluated CTGF expression by TMA. Furthermore, Ki67, a well-known proliferation marker, was also tested. We found significant positive correlations between YAP and CTGF, between YAP and Ki67, between O-GlcNAc and CTGF, and between O-GlcNAc and Ki67 (Fig. 1a and Supplementary Fig. 1a). However, TEAD expression was not correlated with either YAP or O-GlcNAc (Fig. 1a and Supplementary Fig. 1a). These data suggest that the YAP/O-GlcNAc correlation might be associated with cell proliferation, and CTGF expression might be associated with YAP/TEAD-dependent transcriptional activity but is not directly correlated with TEAD expression.

Higher levels of YAP and O-GlcNAcylation were also found in liver cancer tissues compared with normal adjacent liver tissues (Fig. 1d,e), indicating their potential involvement in liver cancer. Similarly, the levels of global O-GlcNAcylation and YAP were much higher in established liver cancer cell lines (SMMC-7721, Bel-7404, Bel-7402, HepG2, Huh7 and SK-Hep1) compared to non-cancerous hepatocyte lines, HL-7702 and THLE-3, and a positive correlation between YAP and global O-GlcNAcylation was also observed in liver cancer cell lines (Fig. 1f). Notably, no significant TEAD expression differences were observed either between clinical liver cancer specimens and their adjacent normal liver tissues or between established liver cancer cell lines and hepatocyte lines (Fig. 1d–f), further demonstrating that O-GlcNAcylation likely exerts its pro-tumorigenic roles specifically via YAP but not by alerting YAP-dependent transcription factors. Because Bel-7402 and SMMC-7721 cells had the highest levels of YAP and global O-GlcNAcylation (Fig. 1f), we chose these two liver cancer cell lines as the main materials for the following study.

Transcriptional activity of TEAD is one of the most important indicators of YAP activity[8,30]. By treating cells with GlcNAc (an activator of O-GlcNAcylation) and PuGNAc (an inhibitor of O-GlcNAcase (OGA), a glycosidase that removes O-GlcNAc modifications), we found that TEAD-dependent transcriptional activities and global O-GlcNAcylation were enhanced compared to the DMSO control (Fig. 2a,b). Moreover, phosphorylation of YAP at Ser127 (p-YAP), a hallmark of YAP inactivation, was reduced, while the total level of YAP and its target, CTGF, was increased by GlcNAc and PuGNAc (Fig. 2b). However, stimulation with GlcNAc alone did not induce an increase in YAP levels, as strongly or significantly as treatment with either PuGNAc alone or combined PuGNAc and GlcNAc treatment (Fig. 2a,b); therefore, in the follow-up experiments, GlcNAc alone was not used to treat cells. The qPCR data showed that the messenger RNA (mRNA) level of CTGF was up-regulated by treatment with GlcNAc and PuGNAc; however, the mRNA level of YAP was little changed (Fig. 2c). These results suggest that enhancing global cellular O-GlcNAcylation can cause a reduction in YAP phosphorylation at Ser127, as well as an increase in the total YAP level, thus increasing YAP activity. These events are independent of YAP transcription.

To exclude the possibility that the results of treatments with GlcNAc at a concentration of 4 mM were due to depletion

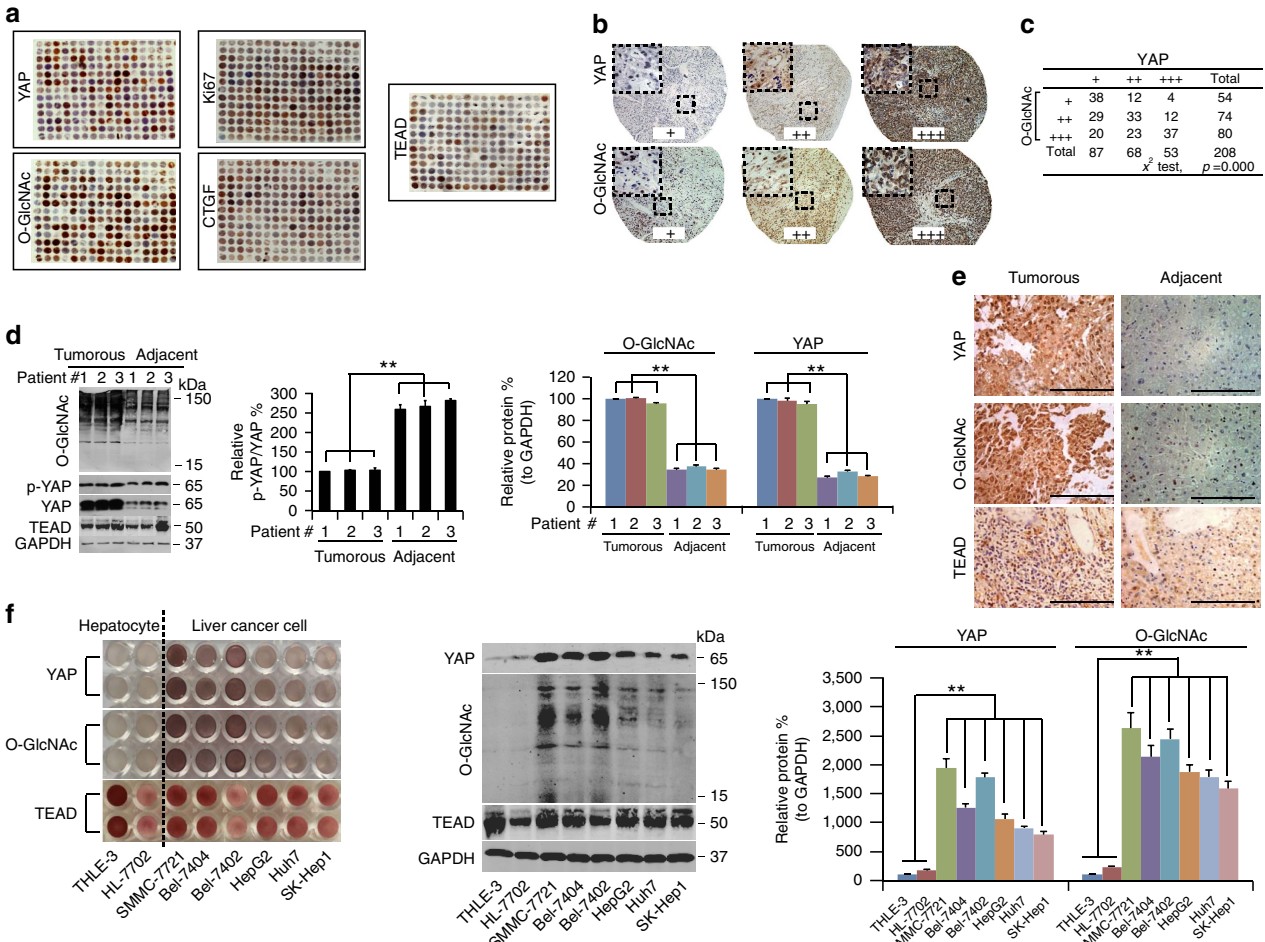

**Figure 1 | YAP expression is associated with global O-GlcNAcylation in liver cancer tissues and established liver cancer cell lines.** (**a–c**) TMA of YAP, Ki67, O-GlcNAc, CTGF and TEAD. IHC images of TMA stained with anti-YAP, anti-Ki67, anti-CTGF, anti-TEAD and anti-O-GlcNAc antibody, (**a**). Enlarged IHC images of YAP and O-GlcNAc staining from the TMA (**b**). The data were analysed using a chi-square test (**c**). (**d–f**) O-GlcNAc and YAP expression in liver cancer and established cell lines. Western blots of O-GlcNAc, TEAD, p-YAP and YAP in liver cancer and adjacent normal liver tissues. The levels of p-YAP were normalized to those of YAP. In addition, the levels of O-GlcNAc and YAP were normalized to those of GAPDH. The levels of the #1 sample of liver cancer were arbitrarily set to 100% (**d**). Representative images of IHC staining of YAP, TEAD and O-GlcNAc in liver cancer and adjacent normal liver tissues. Scale bar, 500 μM. The liver cancer and normal liver tissue samples are paired (**e**). YAP and TEAD expression and O-GlcNAc levels in established hepatocyte (THLE-3 and HL-7702) and the indicated liver cancer cell lines detected by IHC and WB, respectively. The levels of YAP and of O-GlcNAc were normalized to those of GAPDH, and the levels of the THLE-3 cells were arbitrarily set to 100% (**f**). The data are shown as the means + s.d. from three independent experiments (including WB). **$P < 0.01$ indicates statistical significance. Data in **c,d,f** were analysed using a $\chi^2$ test, two-way and one-way ANOVA tests, respectively.

of intracellular ATP in cells, which is consumed in the phosphorylation of free GlcNAc, we examined ATP levels when cells were treated with increasing concentrations of GlcNAc. We found that the ATP levels were not significantly reduced until GlcNAc reached a concentration of 8 mM (Supplementary Fig. 1b), suggesting that ATP is not a major factor affecting GlcNAc-induced phenotypes when the concentration of GlcNAc is below 8 mM.

Because OGT is the only known endogenous enzyme that catalyses O-GlcNAcylation[31,32], we examined the effect of OGT overexpression. In a dose-dependent manner, OGT overexpression elevated TEAD-dependent transcriptional activity (Fig. 2d). Overexpression of OGT also led to decreased phosphorylation of YAP, and an increase in total YAP and global intracellular O-GlcNAcylation in both Bel-7402 and SMMC-7721 cells (Fig. 2d). By contrast, knockdown of OGT led to suppression of TEAD activity, induction of p-YAP, and reduction in total YAP and global intracellular O-GlcNAcylation (Fig. 2e). These results further

support our finding that O-GlcNAcylation positively regulates YAP function.

Activation of YAP leads to an enhanced interaction between YAP and its dependent transcription factors[8,30]. We co-transfected YAP-FLAG with TEAD4-Myc or CREB-HA into Bel-7402 cells and found that OGT overexpression led to increased YAP-FLAG levels in the immunoprecipitates (IPs) pulled down by anti-Myc antibodies or anti-HA antibodies (Supplementary Fig. 1c), providing further evidence that O-GlcNAcylation is capable of stimulating YAP activity.

**O-GlcNAcylation stabilizes YAP via inhibition of βTrcP.** We subsequently investigated how O-GlcNAcylation regulates the YAP expression level. We excluded the possibility that O-GlcNAcylation regulates the mRNA levels of YAP (Fig. 2c). In addition, via cycloheximide (CHX) chase experiments, we found that treatment with PuGNAc and GlcNAc prolonged the half-life of YAP protein

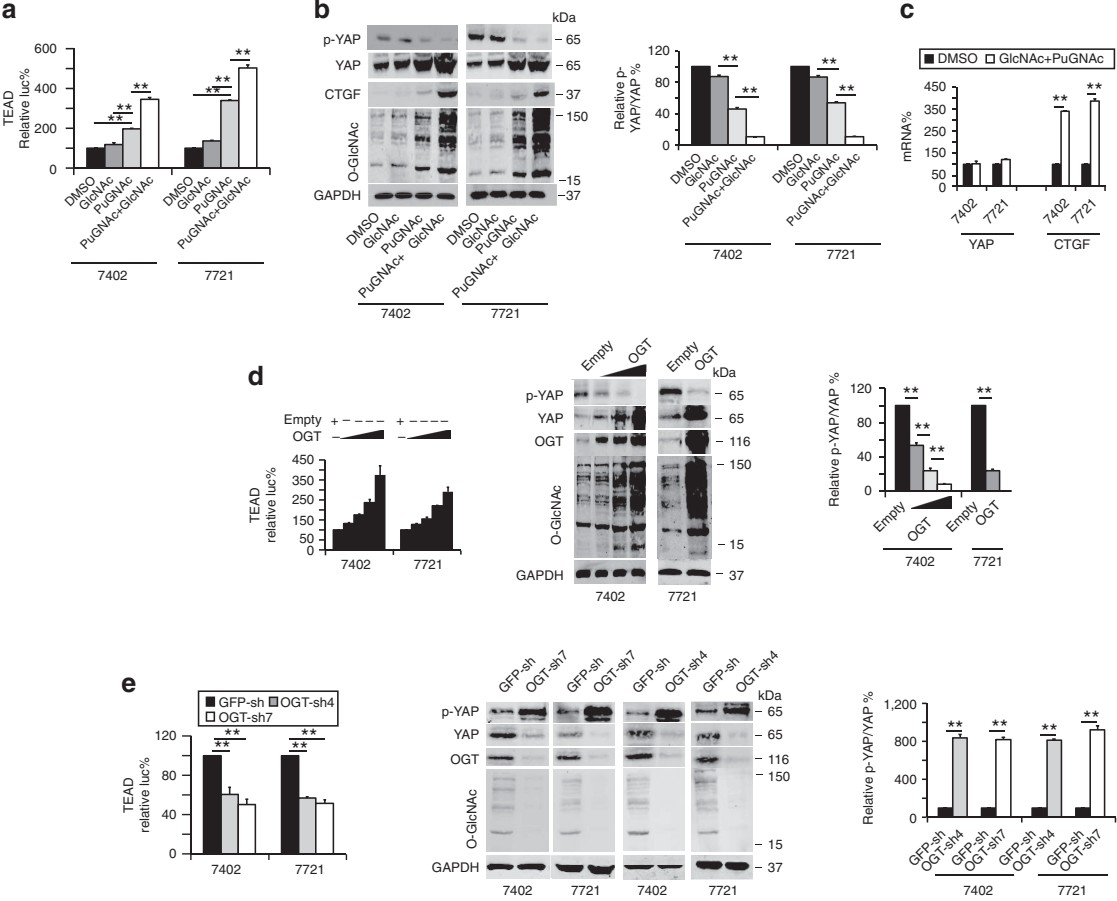

**Figure 2 | O-GlcNAcylation stimulates YAP-dependent transcriptional activity.** (**a**) Luciferase activity from a pUAS-Luc/TEAD-Gal4 system in Bel-7402 and SMMC-7721 cells was measured following treatment with DMSO, GlcNAc (4 mM), and PuGNAc (25 μM) with or without GlcNAc (4 mM) for 24 h. (**b**) Representative western blots of the indicated proteins in Bel-7402 and SMMC-7721 cells treated with DMSO, GlcNAc and PuGNAc with or without GlcNAc for 24 h. The levels of p-YAP were normalized to those of YAP, and the groups treated with DMSO were arbitrarily set to 100%. (**c**) The mRNA levels of YAP and CTGF in Bel-7402 and SMMC-7721 cells treated with DMSO or PuGNAc combined with GlcNAc for 24 h. The mRNA levels were measured by qPCR. (**d**) Overexpression of OGT stimulated YAP activity and expression, as measured by a luciferase-based assay using a pUAS-Luc/TEAD-Gal4 system in Bel-7402 or SMMC-7721 cells transfected with 1–4 μg of OGT-expressing plasmids (left panel). The middle panel shows the representative WB images of p-YAP, YAP, OGT, O-GlcNAc and GAPDH in Bel-7402 (transfected with 1–4 μg of OGT-expressing plasmids) and SMMC-7721 cells (transfected with 4 μg of OGT-expressing plasmids). The right panel shows the ratios between p-YAP and YAP protein levels, and the cells treated with empty vectors were arbitrarily set to 100%. (**e**) Knockdown of OGT suppressed YAP activity and expression in Bel-7402 and SMMC-7721 cells, as measured by a luciferase-based assay using a pUAS-Luc/TEAD-Gal4 system (left) and western blot analysis (middle and right). The levels of p-YAP were normalized to those of YAP, and the cells infected with GFP-sh were arbitrarily set to 100% (right). The data are shown as the means + s.d. from three independent experiments (including WB). \*\*$P < 0.01$ indicates statistical significance. The data from **c,e** (right panel) were analysed by a Student's $t$-test, and data from **a,b,d,e** (left panel) were analysed by one-way ANOVA.

(Fig. 3a). Similarly, overexpression of OGT protected YAP from degradation (Supplementary Fig. 2a), whereas knocking down OGT facilitated YAP degradation (Fig. 3b).

βTrCP has been reported to be a ubiquitin E3 ligase that mediates YAP degradation[13]. Therefore, we investigated whether stimulation of O-GlcNAcylation inhibits YAP degradation through βTrCP. Indeed, overexpression of βTrCP led to a significant reduction in YAP, which could be dose-dependently reversed by simultaneous overexpression of OGT (Fig. 3c). To test whether stimulation of O-GlcNAcylation by PuGNAc and GlcNAc prevents the βTrCP interaction with YAP, we performed reciprocal co-IP experiments and found that βTrCP was dissociated from IPs pulled down by anti-YAP antibodies (Fig. 3d) and *vice versa* (Supplementary Fig. 2b). Moreover, overexpression of OGT suppressed the βTrCP interaction with YAP (Supplementary Fig. 2c), while knockdown of

OGT had the opposite effect (Fig. 3e). These results demonstrate that stimulation of O-GlcNAcylation is capable of enhancing YAP stability by preventing its interaction with the E3 ligase, βTrCP.

**O-GlcNAcylation promotes tumorigenesis via YAP.** The observation that the levels of cellular O-GlcNAcylation were elevated in liver cancer prompted us to investigate whether O-GlcNAcylation has pro-tumorigenic functions. We found that knockdown of OGT resulted in a significant reduction in YAP and O-GlcNAcylation levels (Fig. 4a), cell proliferation (Fig. 4b) and colony formation capacity (Fig. 4c) but an increase in p-YAP levels (Fig. 4a) and Caspase 3/7 activity, a hallmark of apoptosis (Fig. 4d), in both Bel-7402 and SMMC-7721 cells. By contrast, YAP overexpression led to an increase in cell

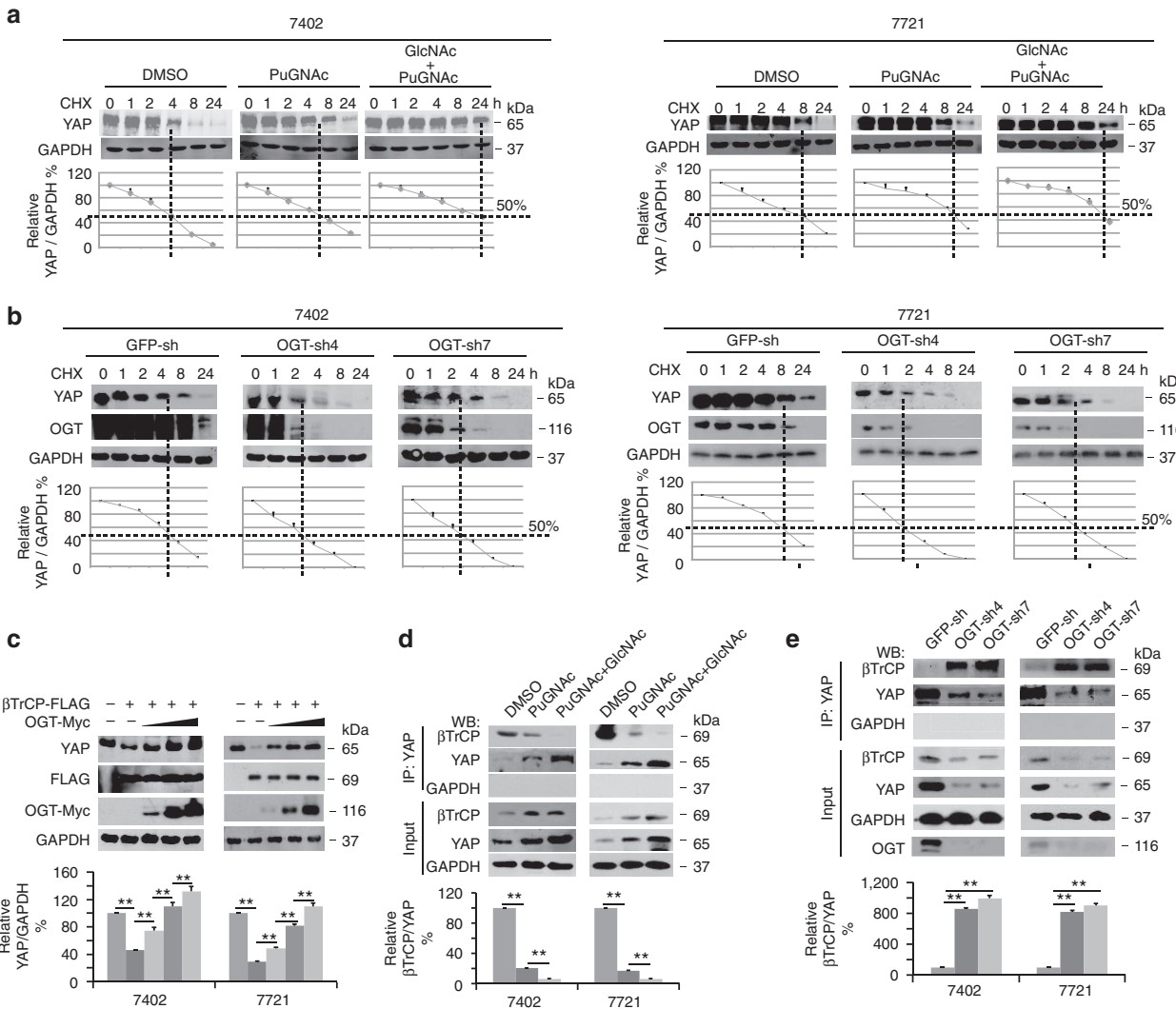

**Figure 3 | O-GlcNAcylation enhances YAP stability by inhibiting ßTrCP.** (**a**,**b**) O-GlcNAcylation increased YAP stability. Protein synthesis was blocked by treatment with CHX (50 µg ml$^{-1}$) for the indicated times. The half-life of endogenous YAP in Bel-7402 and SMMC-7721 cells treated with DMSO or PuGNAc (25 µM) with or without GlcNAc (4 mM) for 24 h (**a**) or in Bel-7402 and SMMC-7721 cells infected with the indicated OGT shRNA (**b**), as measured by western blot analysis. The levels of YAP were normalized to those of GAPDH, and the 0 h points were arbitrarily set to 100%. (**c**) OGT reversed ßTrCP-induced YAP degradation. YAP expression was examined in Bel-7402 and SMMC-7721 cells with ßTrCP overexpressed in the presence or absence of transfection with an increasing amount of OGT-Myc plasmid (transfected with 1–3 µg of OGT-Myc-expressing plasmids). The levels of YAP were normalized to those of GAPDH, and the data from the untreated group (lane 1) were arbitrarily set to 100%. (**d**) Stimulation of O-GlcNAcylation blocked the interaction between ßTrCP and YAP. Bel-7402 and SMMC-7721 cells were treated with DMSO or PuGNAc with or without GlcNAc for 24 h. YAP was immunoprecipitated by anti-YAP antibodies, and the association with ßTrCP was detected by anti-ßTrCP antibodies. The levels of ßTrCP were normalized to those of YAP in the IP samples, and the data from cells treated with DMSO were arbitrarily set to 100%. (**e**) OGT inhibited interactions between ßTrCP and YAP. Bel-7402 and SMMC-7721 cells were infected with the indicated OGT shRNA. YAP was immunoprecipitated by anti-YAP antibodies, and the association with ßTrCP was detected by anti-ßTrCP antibodies. The levels of ßTrCP were normalized to those of YAP in the IP samples, and the data from GFP-sh were arbitrarily set to 100%. Representative images are shown, and the data are expressed as the means + s.d. from three independent experiments. **$P < 0.01$ indicates statistical significance. The data from **c**–**e** were analysed by one-way ANOVA.

proliferation and colony formation capacity, but a decrease in Caspase 3/7 activity. However, its contributions to these transformative phenotypes were not significant (Fig. 4a–d). These results might be because YAP is already overexpressed or over-activated in liver cancer cells[7,33], and therefore, the effects caused by addition of extra-YAP might not be obvious. However, impaired YAP (either p-YAP or total-YAP), global O-GlcNAcylation levels and transformative phenotypes could be reversed by simultaneous overexpression of YAP (Fig. 4a–d). To further confirm the specific contribution of YAP in O-GlcNAcylation-maintained transformative phenotypes, we

selected c-Fos, which is also an oncoprotein that is overexpressed in liver cancer[34], as a control. We observed that knockdown of OGT had no significant effects on c-Fos (Supplementary Fig. 3a). Furthermore, the impaired transformative phenotypes caused by knocking OGT down could not be reversed by simultaneous overexpression of c-Fos (Supplementary Fig. 3b–d). Therefore, we confirmed that O-GlcNAcylation maintains transformative phenotypes in liver cancer cells, likely specifically through YAP O-GlcNAcylation.

Furthermore, the levels of YAP and O-GlcNAcylation were reduced by OGT knockdown, whereas they were partially

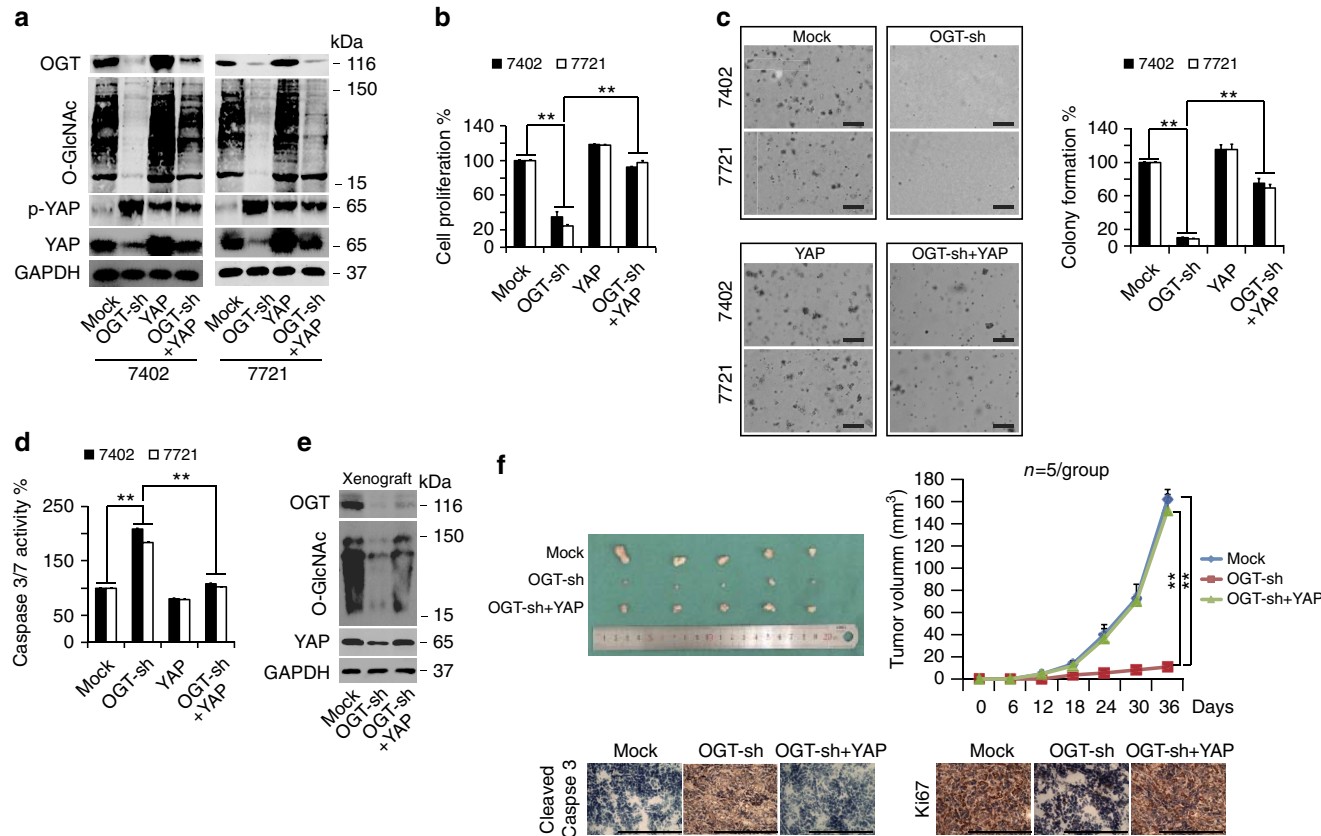

**Figure 4 | O-GlcNAcylation of YAP stimulates tumorigenesis.** (**a–d**) Ectopic expression of YAP rescued the diminished transformative phenotypes caused by OGT knockdown *in vitro*. The protein expression of OGT, O-GlcNAc, p-YAP and YAP in Bel-7402 and SMMC-7721 cells was measured by WB (**a**). Cell proliferation, colony formation capacity and Caspase 3/7 activities were measured with an MTT-based assay (**b**), soft agar colony formation assay (**c**) and Caspase 3/7 Glo luciferase assay (**d**), respectively, in Bel-7402 and SMMC-7721 cells under different treatments as indicated. Scale bar, 500 μm. (**e,f**) Ectopic expression of YAP rescued the xenograft tumour growth inhibited by OGT silencing *in vivo*. The protein expression of OGT, O-GlcNAc and YAP in xenografts was measured by WB (**e**). Tumour volumes were monitored for 36 days after subcutaneous injection of Bel-7402 cells into nude mice; n = 5 per group (**f**). Representative IHC images of cleaved Caspase 3 and Ki67 staining in the xenografts formed in nude mice. Scale bar, 500 μm. Representative images from three independent experiments are shown (**a–e**). \*\*P < 0.01 indicates statistical significance. The data from **b–d,f** were analysed by one-way and two-way ANOVA, respectively.

reversed by simultaneous overexpression of YAP in the xenografts generated by Bel-7402 cells (Fig. 4e). We also found that the growth rate of xenografts with OGT knocked down was much slower than that of the control xenografts, and reduced tumour growth could also be reversed by overexpression of YAP (Fig. 4f). Further IHC analysis of xenograft tissues revealed that OGT knockdown stimulated Caspase 3 cleavage (Fig. 4f) but reduced the expression of Ki67 (Fig. 4f). Similarly, YAP overexpression reversed these OGT knockdown effects (Fig. 4f). Collectively, these *in vitro* and *in vivo* results demonstrate that stimulation of cellular O-GlcNAcylation can promote liver tumorigenesis in a YAP-dependent manner.

**YAP O-GlcNAcylation at Thr241 is required for tumorigenesis.** Can YAP be O-GlcNAcylated in cells? O-GlcNAcylated proteins and OGT were observed in the IPs pulled down by anti-YAP antibodies; conversely, YAP and OGT were detected in the IPs pulled down by anti-O-GlcNAc antibodies, similar to c-Myc, an oncoprotein that has been reported to be O-GlcNAcylated[5] (Supplementary Fig. 4a–b). The direct interactions between YAP and OGT were further confirmed by performing protein ligation assays (PLAs) using a Duolink kit (Supplementary Fig. 4c). Enzymatic labelling of O-GlcNAc sites

using HRP-labelled Streptavidin and anti-TAMRA antibodies revealed that YAP can be directly O-GlcNAcylated (Fig. 5a). Furthermore, *in vitro* O-GlcNAcylation experiments performed by mixing purified YAP and OGT proteins with UDP-GlcNAc resulted in potent O-GlcNAcylation of YAP (Fig. 5a). In liver cancer cells, treatment with PuGNAc and GlcNAc enhanced YAP O-GlcNAcylation (Supplementary Fig. 4d), and overexpression of OGT resulted in a similar enhancement of YAP O-GlcNAcylation, while knockdown of OGT led to suppression of YAP O-GlcNAcylation (Supplementary Fig. 4e). These results suggested that YAP is O-GlcNAcylated by OGT in liver cancer cells.

To exclude the 'O-GlcNAc independent' background that may be generated by the anti-O-GlcNAc antibody (RL2), and to demonstrate the specificity of the effects on the O-GlcNAcylation of YAP, we maintained cells with O-GlcNAc at either low (100 μM) or high (600 μM) concentrations before and during further treatments. We used the RL2 antibody and found that the effects of PuGNAc with or without additional GlcNAc, overexpression or knockdown of OGT on the O-GlcNAcylation of YAP were similar between the two groups (Supplementary Fig. 4f–h). Furthermore, primary hepatocytes from mouse liver were also cultured in low (100 μM) or high (600 μM) concentrations of GlcNAc before and during further treatments

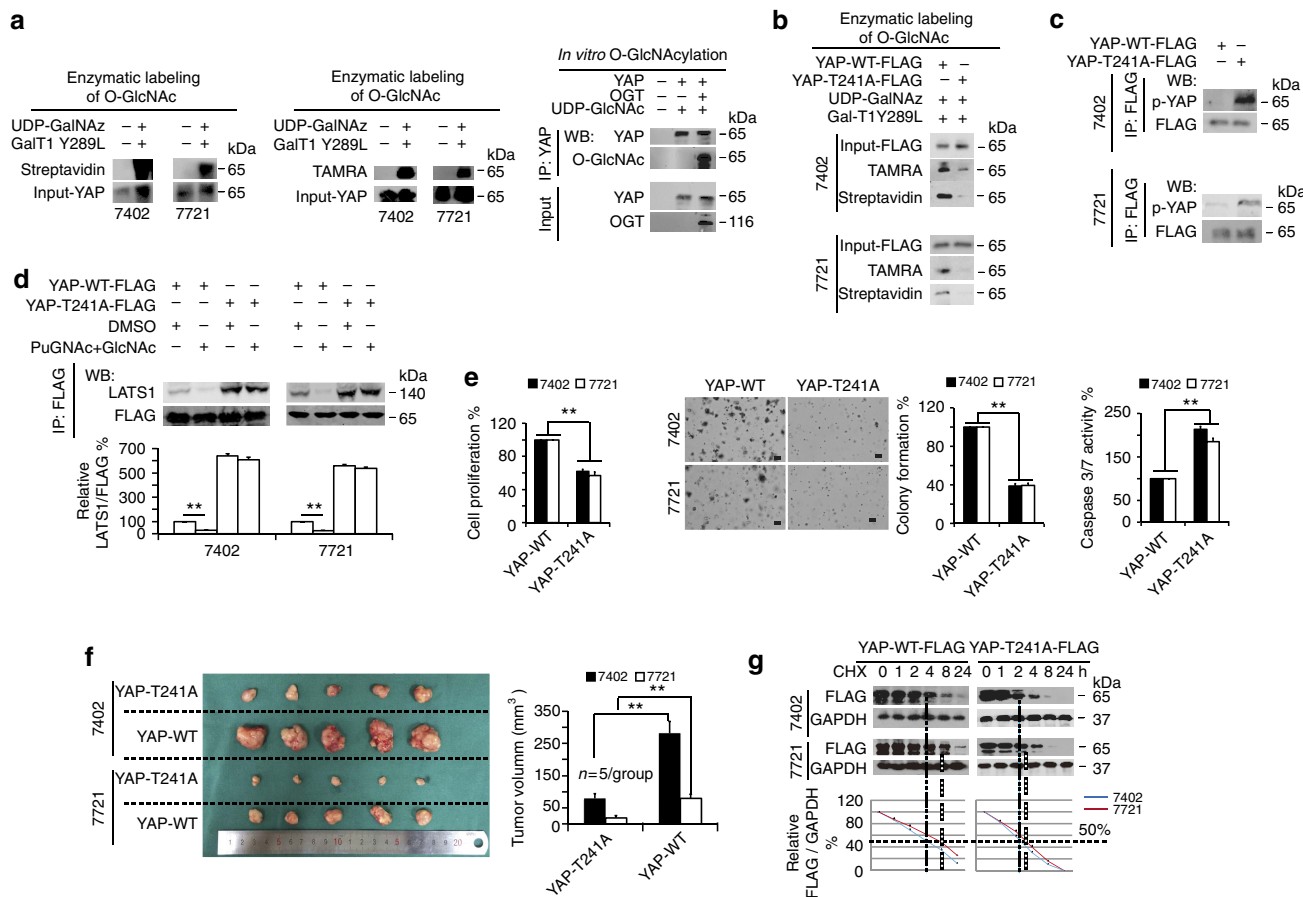

**Figure 5 | YAP is O-GlcNAcylated by OGT at Thr241.** (**a**) YAP O-GlcNAcylation was measured via HRP-labelled Streptavidin (left) and anti-TAMRA antibodies (middle) in Bel-7402 and SMMC-7721 cells. *In vitro* O-GlcNAcylation of YAP was measured by incubating purified YAP and OGT proteins with UDP-GlcNAc at 37 °C for 90 min before immunoprecipitation with anti-YAP antibodies, followed by western blot using an anti-O-GlcNAc antibody (right). (**b**) Enzymatic labelling of O-GlcNAc in WT and T241A YAP-FLAG proteins as analysed by anti-TAMRA antibodies and HRP-labelled Streptavidin in Bel-7402 and SMMC-7721 cells. (**c**) YAP-FLAG was immunoprecipitated by an anti-FLAG antibody in Bel-7402 and SMMC-7721 cells transfected with WT or T241A YAP-FLAG-expressing plasmids. Phosphorylation of YAP was detected using an anti-p-YAP antibody. (**d**) Mutation of Thr241 induced LATS1-YAP binding under stimulation of O-GlcNAcylation by combined treatment of GlcNAc (4 mM) and PuGNAc (25 μM). Expression plasmids for YAP-FLAG, WT or T241A were transfected into Bel-7402 and SMMC-7721 cells, and cells were treated as indicated for 24 h before analysis. YAP-FLAG was immunoprecipitated with anti-FLAG antibodies, and the association with LATS1 was measured using anti-LATS1 antibodies. (**e**) Mutation of Thr241 reduced YAP-promoted transformative phenotypes *in vitro*. Cell proliferation, colony formation capacity and Caspase 3/7 activities were measured with MTT-based assays, soft agar colony formation assays and Caspase 3/7 Glo luciferase assays, respectively, in Bel-7402 and SMMC-7721 cells expressing WT or T241A YAP-FLAG. Scale bar, 500 μm. (**f**) Mutation of Thr241 diminished YAP-promoted xenograft tumour formation *in vivo*. The tumour size was measured 72 days after injection of Bel-7402 or SMMC-7721 cells into nude mice. (**g**) Mutation of Thr241 reduced the half-life of YAP. YAP-FLAG (as indicated) was transfected into Bel-7402 and SMMC-7721 cells, and cells were harvested at the indicated time points after addition of CHX (50 μg ml$^{-1}$). The expression of YAP-FLAG was normalized to that of GAPDH, and the '0 h' point was arbitrarily set to 100%. Representative images are shown, and the data are expressed as the means + s.d. from three independent experiments. \*\*$P < 0.01$ indicates statistical significance. The data from **d**–**f** were analysed by Student's *t*-test.

with PuGNAc, with or without additional GlcNAc, and we found that O-GlcNAcylation of YAP detected by the RL2 antibody was similar between the two groups (Supplementary Fig. 4i). These results suggest that the effects of O-GlcNAcylation on YAP detected by the RL2 antibodies were specific.

Next, mass spectrometry (MS) analysis was performed, and we identified a GlcNAc-modified peptide of YAP (Supplementary Figs 5–7). Although Thr241 was predicted as a potential GlcNAc site by a Mascot search, and interestingly, this site is located within the second WW domain of the YAP1-2 isoform, we could not exclude the possibility of GlcNAc modifications on other serine or threonine sites. Therefore, we also performed point mutation experiments for other serine and threonine sites within this peptide, and observed that when Ser208, Thr213, Thr216, Ser217, Ser227 or

Ser229 were mutated, the levels of O-GlcNAc-modified YAP were almost unchanged compared to wild type (WT) YAP. However, when Thr241 was mutated, the levels of O-GlcNAc-modified YAP were significantly decreased (Supplementary Fig. 8a), suggesting that Thr241 is the only GlcNAc site within this peptide. Importantly, this site is conserved across species (Supplementary Fig. 8b). When Thr241 was replaced by an alanine (T241A mutant), YAP O-GlcNAcylation was significantly reduced compared to WT YAP, as analysed by enzymatic labelling of O-GlcNAc sites using anti-TAMRA antibodies and HRP-labelled Streptavidin (Fig. 5b), further indicating that Thr241 is the major O-GlcNAcylation site of YAP.

Because stimulation of cellular O-GlcNAcylation can reduce phosphorylation of YAP, we hypothesized that O-GlcNAcylation

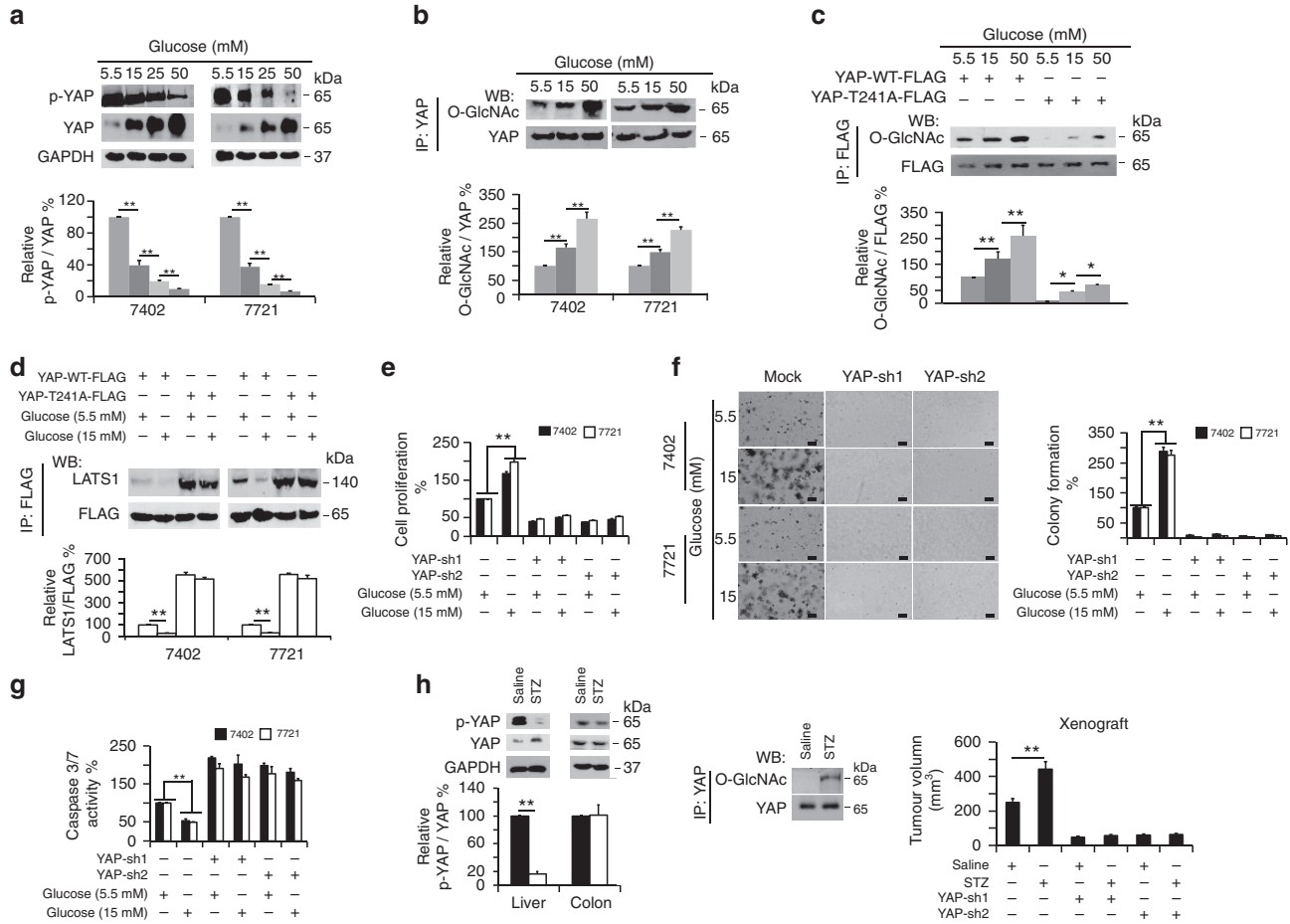

**Figure 6 | YAP plays a critical role in high-glucose-stimulated liver tumorigenesis.** (**a**) Glucose inhibited YAP phosphorylation while stimulating YAP expression. Bel-7402 and SMMC-7721 cells were cultured in media containing the indicated concentrations of glucose for 24 h. YAP expression levels and phosphorylation were then measured by western blot. The levels of p-YAP were normalized to those of YAP, and the data from cells cultured in 5.5 mM glucose were arbitrarily set to 100%. (**b,c**) Glucose stimulated O-GlcNAcylation of YAP at Thr241. Bel-7402 and SMMC-7721 cells were cultured in media containing the indicated concentrations of glucose (**b**). Bel-7402 cells were transfected with WT and T241A YAP-FLAG, and cells were treated with glucose at the indicated final concentration for 24 h (**c**). Endogenous YAP or exogenous WT or T241A YAP-FLAG was immunoprecipitated with anti-YAP or anti-FLAG antibodies, and O-GlcNAcylation was measured by western blot using an anti-O-GlcNAc antibody. The levels of O-GlcNAcylation of endogenous YAP or exogenous YAP-FLAG were normalized to those of total YAP or YAP-FLAG, and data from cells cultured in 5.5 mM glucose were arbitrarily set to 100%. (**d**) Mutation of Thr241 induced LATS1-YAP binding under glucose stimulation. Expression plasmids of YAP-FLAG, WT or T241A were transfected into Bel-7402 and SMMC-7721 cells, and the cells were treated with the indicated final concentration of glucose for 24 h before analysis. YAP-FLAG was immunoprecipitated with anti-FLAG antibodies, and the association with LATS1 was measured by anti-LATS1 antibodies. (**e–g**) Knockdown of YAP abolished high-glucose-induced tumorigenesis. Cell proliferation, colony formation capacity and Caspase 3/7 activities were measured by MTT-based assays (**e**), soft agar colony formation assays (**f**) and Caspase 3/7 Glo luciferase assays (**g**) in Bel-7402 and SMMC-7721 cells under the indicated treatments. Two shRNA sequences targeting YAP, sh1 and sh2, were used as indicated. Scale bar, 500 μm. (**h**) YAP and p-YAP were detected in the liver and colon of saline- and STZ-treated mice by western blot (left). O-GlcNAcylation of YAP in the liver from saline- and STZ-treated mice. YAP was immunoprecipitated with an anti-YAP antibody, and O-GlcNAcylation was detected with anti-O-GlcNAc antibodies using western blotting (middle). Silencing of YAP inhibited STZ-induced xenograft growth. Tumor volumes were monitored for 36 days after subcutaneous injection of different Bel-7402 cells as indicated; $n = 5$ per group (right). Representative images are shown, and the data are expressed as the means + s.d. from three independent experiments (**a–g**). *$P < 0.05$ and **$P < 0.01$ indicate statistical significance. The data from **a–c** were analysed by one-way ANOVA, and the data from **d–h** were analysed by Student's $t$-test.

of YAP at Thr241 is responsible for this effect. To test this possibility, exogenous YAP-FLAG, WT or T241A mutant was expressed in Bel-7402 and SMMC-7721 cells. We found that phosphorylation of YAP at Ser127 was significantly higher in the T241A mutant compared to WT YAP (Fig. 5c), suggesting that occupancy of O-GlcNAc at Thr241 antagonizes phosphorylation at Ser127. LATS is one type of kinase that can directly phosphorylate YAP at the Ser127 site[14,35]. Therefore, we hypothesized that O-GlcNAcylation at Thr241 affects LATS binding to YAP. To address this, we stimulated O-GlcNAcylation

and investigated whether the YAP-LATS interaction is affected in the presence or absence of mutation at Thr241 in liver cancer cells. We found that stimulation of O-GlcNAcylation by combined treatment with PuGNAc and GlcNAc inhibited LATS1 binding to WT-YAP. However, this inhibitory effect was not significant when Thr241 was mutated. We also noted that mutation of Thr241 significantly increased YAP-LATS1 binding at the basal level (Fig. 5d). Moreover, we excluded the possibility that stimulation of O-GlcNAcylation influences phosphorylation of LATS1 (Supplementary Fig. 8c), suggesting that the reduction

of YAP-LATS1 binding is not due to modification of LATS and instead is likely due to changes in YAP, especially modification at Thr241.

To test whether the Thr241 site is important for YAP-dependent tumorigenesis, we expressed T241A and WT YAP, respectively, in Bel-7402 and SMMC-7721 cells. We found that mutation of Thr241 significantly impaired cell proliferation and colony formation capacity (Fig. 5e). In contrast, Caspase 3/7 activity in cells expressing T241A was much higher than in cells expressing WT YAP (Fig. 5e). Importantly, *in vivo* xenograft experiments also demonstrated that T241A YAP had less pro-tumorigenic properties than WT YAP (Fig. 5f). Moreover, we found the T241A YAP had a reduced half-life compared with WT YAP (Fig. 5g), suggesting that Thr241 is also critical for maintenance of YAP protein stability. All these results demonstrate an essential role of Thr241 in YAP-dependent tumorigenesis.

Notably, higher O-GlcNAcylation levels of YAP were detected in both liver cancer tissues and established cell lines (SMMC-7721, Bel-7404, Bel-7402, Huh7, HepG2 and SK-Hep1) compared to those in neighbouring normal liver tissues and hepatocyte lines HL-7702 and THLE-3 (Supplementary Fig. 8d). We also developed an antibody that specifically recognizes O-GlcNAcylation of YAP at Thr241 (we named this antibody anti-O-T241-YAP) and used this antibody to directly evaluate O-GlcNAcylation of YAP at Thr241 in clinical liver cancer specimens and adjacent normal liver tissues. We found that O-GlcNAcylation of YAP at Thr241 (O-T241-YAP) was significantly higher in tumorous tissues compared to paired adjacent normal tissues in the liver (Supplementary Fig. 8d). We also examined O-T241-YAP in the same tissue microarray that was used in Fig. 1a, and found that O-T241-YAP was significantly associated with the levels of total-YAP, global O-GlcNAcylation, CTGF and Ki67 (Supplementary Fig. 8e), further suggesting that global O-GlcNAcylation might reinforce YAP expression via O-GlcNAcylation of YAP at Thr241, and O-GlcNAcylation of YAP at Thr241 might be involved in the regulation of TEAD-dependent transcription and cell proliferation in liver cancer cells.

Moreover, we performed IF experiments in clinical liver cancer specimens and found that, similar to the localization of YAP, O-GlcNAcylation of proteins occurred mainly in the nucleus, and higher expression of YAP was correlated with higher global O-GlcNAcylation (Supplementary Fig. 8f). Furthermore, higher expression of O-T241-YAP was found in liver cancer tissues with higher total-YAP expression, and O-T241-YAP was also found mainly in the nucleus (Supplementary Fig. 8f). These data further suggest that the nuclear portion of YAP can be O-GlcNAcylated at Thr241.

**YAP mediates high-glucose-stimulated liver tumorigenesis**. Uncontrolled hyperglycaemia is an important risk factor for liver cancer[25]. YAP is also a stimulator of liver tumorigenesis[7,36]. However, the link between high glucose and YAP during liver tumorigenesis remains unclear. To examine their potential relationship, we treated Bel-7402 and SMMC-7721 cells with increasing concentrations of glucose. We found that both YAP expression (Fig. 6a and Supplementary Fig. 9a) and TEAD luciferase activity (Supplementary Fig. 9b) were elevated by increasing concentrations of glucose. Moreover, nuclear accumulation of YAP stimulated by increasing concentrations of glucose were also detected in immunofluorescence and WB experiments (Supplementary Fig. 9a). Furthermore, interaction of YAP with both TEAD and CREB was enhanced by increasing concentrations of glucose, as shown by co-IP experiments

(Supplementary Fig. 9c). These results demonstrate that glucose is capable of stimulating YAP expression and function. Moreover, we found that intracellular Glucose-6-phosphate (G-6-P) and glucose were up-regulated upon glucose stimulation in a dose-dependent manner (Supplementary Fig. 9d), suggesting that our assays measured intracellular phosphorylated glucose and total glucose with the same accuracy as free glucose.

Interestingly, glucose not only increased the total levels of YAP but also reduced the phosphorylation of YAP (Fig. 6a), reminiscent of the effect of the stimulation of cellular O-GlcNA-cylation. We also found that glucose led to a dose-dependent increase of endogenous YAP O-GlcNAcylation in both Bel-7402 and SMMC-7721 cells (Fig. 6b). Notably, glucose had a more marked impact on O-GlcNAcylation of WT YAP compared to the T241A mutant (Fig. 6c). Like the effects of O-GlcNAcylation (Fig. 5d), high glucose (HG, 15 mM) suppressed LATS1 binding to WT YAP compared to normal glucose (NG, 5.5 mM) conditions. However, these effects were not as significant in experiments with the T241A mutant YAP (Fig. 6d). Moreover, mutation of Thr241 induced YAP-LATS1 binding at the basal level (Fig. 6d). To exclude the possibility that HG-suppressed YAP-LATS binding occurs by affecting the phosphorylation of LATS, we tested p-LATS1 and total-LATS1 in the presence or absence of HG. We found that compared to NG conditions, HG had no effect on the phosphorylation of LATS1 (Supplementary Fig. 9e). These data demonstrate that glucose might exert its roles on YAP through O-GlcNAcylation of this protein at Thr241.

To determine the effect of YAP on the transformative phenotypes of liver cancer cells under various glucose concentrations, we measured cell proliferation, colony formation capacity, and apoptosis. Compared to NG, HG was capable of inducing a significant increase in cell proliferation and colony formation capacity, as well as a decrease in apoptosis (Fig. 6e–g). However, in YAP-depleted liver cancer cells, HG was unable to induce enhanced transformative phenotypes (Fig. 6e–g), indicating that YAP is required for HG-stimulated liver tumorigenesis.

To exclude the possibility that glucose alters YAP as a result of osmotic stress, we treated cells with NaCl at a concentration of 50 mM to achieve the same osmotic level as that of 50 mM glucose. Compared to NaCl, stimulation with glucose decreased the p-YAP level (Supplementary Fig. 9f) and Caspase3/7 activity (Supplementary Fig. 9g), while it increased the total YAP levels (Supplementary Fig. 9f), O-GlcNAcylation of YAP (Supplementary Fig. 9h), cell proliferation (Supplementary Fig. 9i) and colony formation capacity (Supplementary Fig. 9j) in both Bel-7402 and SMMC-7721 cells. Moreover, by treating cells with the same concentrations of D-glucose (the functional form of glucose in the human body, and this type of glucose is used throughout the whole study) or L-glucose (the enantiomer of D-glucose, which cannot be used by living organisms as a source of energy), we found that only high concentrations of D-glucose could reduce p-YAP levels (Supplementary Fig. 9k), while inducing total-YAP levels (Supplementary Fig. 9k) and O-GlcNAcylation of YAP (Supplementary Fig. 9l). Therefore, we propose that the HG effects on YAP are independent of osmotic stress.

Subsequently, we performed a series of *in vivo* analyses using streptozocin (STZ)-induced, insulin-deficient diabetic mice, whose serum glucose was elevated, while insulin was decreased compared to the saline-treated control mice (Supplementary Fig. 9m). STZ treatment induced an increase in YAP expression and a concurrent decrease in YAP phosphorylation in the liver but not in the colon of mice (Fig. 6h and Supplementary Fig. 9n). Moreover, there were more dividing cells detected in the liver but not in the colon of STZ-treated mice compared to the control (Supplementary Fig. 9n), suggesting that HG might promote cell

proliferation, possibly restricted to the liver. Furthermore, O-GlcNAcylation of YAP was significantly elevated in the liver upon STZ treatment (Fig. 6h). In addition, STZ could induce significant *in vivo* growth of xenograft tumours (Bel-7402 cells as the donors), which was abolished by YAP depletion (Fig. 6h), further demonstrating the important roles of YAP in HG-stimulated liver tumorigenesis.

In addition, we found that total-YAP and global O-GlcNAcylation levels were higher in the liver cancer tissues from patients with diabetes compared to those without diabetes. By contrast, the p-YAP levels were lower in the liver cancer specimens from patients with diabetes compared to those without (Supplementary Fig. 9o). These results establish a close relationship between YAP and hyperglycaemia in liver tumorigenesis.

**YAP reversely regulates cellular O-GlcNAcylation**. The HBP controls metabolic flux and subsequent O-GlcNAcylation[37]. We have described above that O-GlcNAcylation of YAP promoted its stability and transcriptional activity. However, whether YAP can in turn control HBP and global O-GlcNAcylation in cells remains unknown. Consistent with such a reciprocal regulatory mechanism, we found several HBP-stimulating metabolites, such as glutamine, acetyl-CoA and fructose-6-P, which were all increased by YAP overexpression and conversely decreased by YAP knockdown (Supplementary Table 1). Furthermore, YAP had the opposite effect on lactic acid, a metabolite that can suppress the HBP (Supplementary Table 1). Glycogen, which is not directly involved in HBP, was not significantly altered by either overexpression or knockdown of YAP (Supplementary Table 1). These results suggest that YAP might directly regulate HBP.

In addition, in experiments using cultured Bel-7402 and SMMC-7721 cells, we found that YAP overexpression caused an increase in intracellular glucose levels and a decrease in the glucose levels in the culture media, whereas YAP knockdown had the opposite effect (Fig. 7a). Moreover, intracellular G-6-P could also be up-regulated by YAP overexpression, and conversely, it was down-regulated by YAP knockdown (Supplementary Fig. 10a). These results suggest that YAP regulates cellular glucose uptake. Notably, OGT also had a similar effect on intracellular glucose and G-6-P levels (Fig. 7a; Supplementary Fig. 10a).

Next, we evaluated the expression of genes involved in the HBP. These genes include OGT, Nudt9, SLC5A3, GLUT1, HK1/2, GUCY1A3, CANT1, GFPT1, GNPNAT1, PGEM1/2/3 and UAP1. Using qPCR, we found that the mRNA levels of only OGT, Nudt9 and SLC5A3 were significantly reduced by YAP knockdown and induced by YAP overexpression (Supplementary Fig. 10b–e; Fig. 7b). The protein levels of OGT, SLC5A3 and Nudt9 were also positively associated with the expression of YAP (Fig. 7b). By testing promoter activities using luciferase-based experiments, we found that the promoter activities of *OGT*, *Nudt9* and *SLC5A3* genes were also positively controlled by YAP (Supplementary Fig. 10f). These results demonstrate that YAP modulates the expression of OGT, Nudt9 and SLC5A3 via a transcription-dependent mechanism.

YAP exerts its transcriptional activity through transcription factors, including TEAD and CREB[7,8,38]. We found that YAP-induced mRNA expression and promoter activities of *OGT*, *Nudt9* and *SLC5A3* genes could be enhanced by simultaneous overexpression of either TEAD or CREB (Fig. 7c), further confirming that the HBP is transcriptionally regulated in a YAP-dependent manner.

Finally, we investigated whether YAP directly controls global O-GlcNAcylation in liver cancer cells. We found that YAP knockdown led to a suppression of overall O-GlcNAcylation in cells, whereas YAP overexpression had the opposite effect (Fig. 7d). Moreover, treatment by either combined GlcNAc and PuGNAc or glucose could induce a significant increase in global O-GlcNAcylation in Bel-7402 cells; however, such effects were ablated in YAP-depleted cells (Fig. 7e), demonstrating that YAP is crucial for the stimulation of cellular O-GlcNAcylation.

**Discussion**

YAP stimulates tumorigenesis in liver cancer[17,39]. O-GlcNAcylation is an important posttranslational modification of proteins and plays pro-oncogenic roles in several types of cancer, including liver cancer[40–44]. O-GlcNAcylation of the Jun proto-oncogene (c-Jun) and Tribbles pseudokinase 2 (TRIB2) has been recently reported to stimulate liver tumorigenesis[45,46]. Interestingly, both c-Jun and TRIB2 are nuclear proteins. In the current study, similar to c-Jun and TRIB2, we also found that both O-GlcNAcylated proteins and O-T241-YAP are primarily expressed in the nucleus. Therefore, we speculate that O-GlcNAcylation is more likely to occur in the nucleus, and this modification may easily enhance the expression and function of nuclear proteins, such as YAP. However, whether and how O-GlcNAcylation tends to occur in the nucleus is still unclear and needs to be further investigated.

As for phosphorylation, another type of posttranslational modification, low-phosphorylation of YAP is closely associated with liver cancer[39]. O-GlcNAc and phosphate often occupies amino acids (a.a.) residues in the same substrate[2]. For example, IRS-1, a highly phosphorylated protein critical to insulin and IGF-1 receptor signalling, is either O-GlcNAcylated or phosphorylated at multiple residues[47,48]. Here, we demonstrate that YAP is O-GlcNAcylated at Thr241, a highly conserved site within the central region of this protein. Our study also reveals that this novel modification of YAP antagonizes its phosphorylation at Ser127. Phosphorylation at Ser127 inactivates YAP by facilitating its degradation by βTrCP[13]. Consequently, Thr241 O-GlcNAcylation of YAP is capable of enhancing YAP-dependent liver tumorigenesis. Furthermore, TEAD and CREB are transcription factors mediating YAP-dependent transcriptional activation in liver cancer cells[8,30,36], and we found that stimulation of O-GlcNAcylation induces both YAP-TEAD and YAP-CREB interactions and the subsequent functions of these transcription factors.

Although, O-GlcNAcylation typically regulates phosphorylation within a few a.a., we reported here that O-GlcNAcylation and phosphorylation are simultaneously present on distant sites of YAP protein, and O-GlcNAcylation of Thr241 alters phosphorylation of Ser127. Notably, YAP is not the only example of this. The residues Ser550, Thr648, Ser654 and Thr317 of the FOXO1 protein are O-GlcNAcylated, while Ser318, Ser322, Ser325 and Ser329 are phosphorylated[49]. Interestingly, the Thr241 O-GlcNAc site is located within the second WW domain of the YAP1 isoform that contains two WW domains (known as YAP1-2). These two WW domains are located at a.a. 174–203 and 233–263. The WW domains are protein–protein interaction modules with two signature tryptophan (W) residues that can bind to ligands containing proline-rich sequences such as PPxY motifs[50–54]. Mutation of the WW domains diminishes the ability of YAP to promote cell proliferation, serum-independent growth and oncogenic transformation[54]. Our study has provided another piece of evidence supporting the importance of the WW domain during YAP-induced tumorigenesis because O-GlcNAcylation of Thr241 within the second WW domain boosts YAP protein stability. Some studies have also reported that LATS1, a kinase that phosphorylates YAP at Ser127, can bind with YAP though the WW domain[55,56]. We found that stimulation of cells, either by O-GlcNAcylation or high glucose, inhibits LATS1 binding to WT-YAP; however, this inhibitory

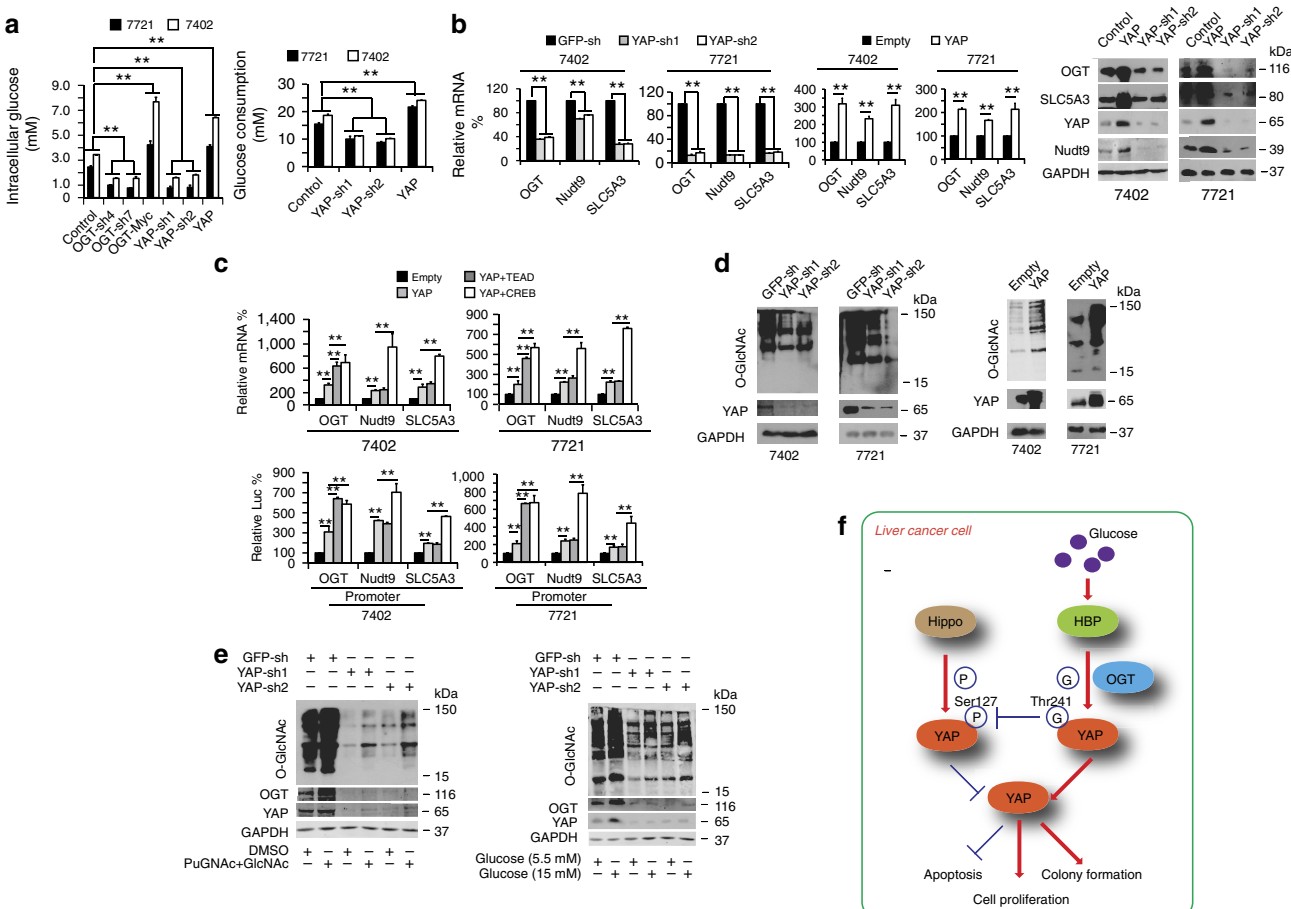

**Figure 7 | YAP reversely stimulates O-GlcNAcylation. (a)** OGT and YAP increased the intracellular glucose level and glucose uptake in Bel-7402 and SMMC-7721 cells. The intracellular glucose levels were measured directly from cell lysates ($1.25 \times 10^6$ cells/100 µl of lysis buffer, left). The glucose consumption levels were calculated as the difference between the initial and the remaining glucose levels in the cell culture media of cells subjected to different treatments as indicated (right). **(b)** OGT, Nudt9 and SLC5A3 were analysed by qPCR or WB in control cells and in Bel-7402 or SMMC-7721 cells with YAP knocked down or overexpressed. **(c)** Bel-7402 or SMMC-7721 cells were transfected with expression plasmids as indicated, and the mRNA levels of OGT, Nudt9 and SLC5A3 were analysed with qPCR. The promoter activities of *OGT*, *Nudt9* and *SLC5A3* genes were measured with a dual luciferase analysis. **(d)** YAP stimulated O-GlcNAcylation. O-GlcNAcylation was analysed by WB in control cells and in Bel-7402 or SMMC-7721 cells with YAP knocked down (left) or overexpressed (right). **(e)** Stimulation of O-GlcNAcylation was dependent on YAP. Control and Bel-7402 cells with YAP knocked down were treated with DMSO, PuGNAc (to a final concentration of 25 µM) with GlcNAc (to a final concentration of 4 mM) for 24 h before being harvested to test the indicated proteins using WB (left). The same proteins were also tested by WB in control cells and Bel-7402 cells with YAP knocked down after treatment with the indicated concentration of glucose for 24 h (right). **(f)** Schematic diagram of a possible mechanism underlying the interaction between O-GlcNAcylation and YAP during tumorigenesis in liver cancer cells. The data represent the means + s.d. from three independent experiments. **$P < 0.01$ indicates statistical significance. The data from **a,b** (left two panels) and **c** were analysed by one-way ANOVA, and the data from **b** (middle two panels) were analysed by Student's *t*-test.

effect was not significant when Thr241 is mutated. These data suggest that O-GlcNAcylation of YAP at Thr241 within the WW domain prevents LATS1 interaction with YAP for further phosphorylation. Interestingly, Li *et al.* reported that phosphorylation of tyrosine188 (Y188) in the YAP1-2 isoform stimulates YAP1-induced cellular transformation. Mutation of Y188 (especially replacement of Y to phenylalanine (F)) leads to a higher affinity of YAP for binding to its upstream negative regulators for cytoplasmic retention[57]. Like Thr241, the Y188 site is also located in the conserved aromatic core of the second WW domain of YAP1. These findings further demonstrate that posttranslational modifications of WW domains may play significant roles in the function of YAP, specifically inducing changes in the ability of YAP to form complexes with other proteins. Whether Y188 phosphorylation and Thr241 O-GlcNAcylation have an impact on each other needs to be

further investigated. The WW domain containing transcription regulator 1 (TAZ), a paralogue of YAP, has a structure similar to that of YAP. Unlike YAP, all the isoforms of TAZ in human cells do not have the second WW domain[58]. Comparatively speaking, YAP is a more prevalent oncoprotein in liver cancer. However, the function of TAZ in liver cancer is limited. We speculate that O-GlcNAcylation on the second WW domain of YAP plays an important role in promoting liver tumorigenesis, and this finding also supports the notion that YAP is more important than TAZ, because TAZ has only one WW domain that might not be O-GlcNAcylated. A series of studies[53,59,60] have also demonstrated that both YAP1-1 and YAP1-2 are present in liver tissue, and YAP1-2 has stronger transactivation activity compared to that of YAP1-1 (ref. 59). Therefore, it is also not difficult to conclude that O-GlcNAcylation of the YAP1-2 proportion at its second WW domain might enhance the YAP

pro-tumorigenic function contributed by both YAP1-1 and YAP1-2.

Data from the current study suggest that Thr241 might not be the only O-GlcNAc site within YAP1. It is common that several O-GlcNAc-modification sites are present in one glycoprotein. We suggest that Thr241 is the most pivotal O-GlcNAc site within YAP. In the MS experiment, the instrument has a certain sensitivity. The expression of O-T241-YAP reached a level that was detectable by MS. However, the expression of other O-GlcNAc-modifications on YAP might not reach the cut-off value of the MS instrument. Moreover, when YAP was mutated on Thr241, the O-GlcNAc-modified level of YAP was significantly decreased. Furthermore, the T241A mutant YAP exhibited reduced protein stability and an impaired capacity to maintain transformative phenotypes compared to WT YAP. Therefore, we believe that although there might be other O-GlcNAc-modified sites within the YAP protein, Thr241 is the most important one.

A high-glucose level is one of the most important risk factors for liver cancer. High glucose is one of the major characteristics of type 2 diabetes. Type 2 diabetes is associated with an increased risk of liver cancer, as numerous studies have reported[61–63]. In one meta-analysis[61], authors found that hepatitis C virus-infected or cirrhotic patients with concomitant type 2 diabetes are more likely to develop liver cancer than those without diabetes. Type 2 diabetes is also reported to be associated with a poor prognosis in liver cancer[62]. Therefore, as a common characteristic of both type 1 and 2 diabetes, high glucose might increase the risk of the occurrence of liver cancer. Moreover, high glucose can lead to the synthesis of excessive UDP-GlcNAc, the direct substrate of O-GlcNAcylation[22]. Here, we have revealed that high glucose is able to induce O-GlcNAcylation and expression and transcriptional activity of YAP, leading to YAP-dependent tumorigenic phenotypes in both cultured liver cancer cells and xenograft models. Therefore, enhanced YAP O-GlcNAcylation contributes to liver tumorigenesis caused by uncontrolled hyperglycaemia. Interestingly, we found that YAP can promote glucose uptake, synthesis of metabolites used in the HBP, and cellular O-GlcNAcylation, thus establishing a positive feedback loop. In addition, YAP has been reported to stimulate PI3K/AKT/mTOR signalling[64,65]. Because activation of PI3K/AKT/mTOR signalling is sufficient to elevate O-GlcNAcylation[40,66], it is important to elucidate in the future the potential interplay between HBP/O-GlcNAcylation signalling and the YAP/PI3K/AKT/mTOR axis.

Taken together, this study has revealed the molecular basis underlying the crucial roles of YAP O-GlcNAcylation in liver tumorigenesis, particularly high-glucose-associated liver tumorigenesis (Fig. 7f). This study indicates that blocking YAP O-GlcNAcylation might be a promising therapeutic strategy for treating high-glucose-associated liver cancer.

## Methods

**Mouse experiments.** HG and control mouse models were constructed by intraperitoneal injection of $40\,mg\,kg^{-1}$ Streptozocin (STZ, dissolved in a solution of 0.1 M citrate acid, pH 4.4) and saline, respectively, into 5-week-old Balb/c male mice (Bikai, Shanghai, China) once a day for four consecutive days. For xenograft mouse experiments, Bel-7402 or SMMC-7721 cells ($5 \times 10^6$ cells) under different treatments were subcutaneously injected into 8-week-old Balb/c male athymic nude mice (Bikai) treated with or without STZ. The tumour size was measured 36 or 72 days after injection, and the tumour volume was calculated as $0.5 \times L \times W^2$, with $L$ indicating length and $W$ indicating width. There were five mice per group in all mouse experiments, and all mouse experiments were performed according to the institutional guidelines of Shanghai Tenth People's Hospital, which has the permission for animal experimentation from the Science and Technology Commission of the Shanghai Municipality.

**Tissue samples.** Tumorous and adjacent liver tissues (Fig. 1d,e and Supplementary Fig. 8d) were acquired at Shanghai Ruijin Hospital under

institutional approvals. Another 12 liver cancer tissues from patients complicated with or without diabetes, as a complicating factor (Supplementary Figs 8f and 9o) were collected from Shanghai Tenth People's Hospital under institutional approvals. Informed written consent was obtained from all the patients.

**Cell culture and vectors.** The liver cancer cell lines Bel-7402 (Cell bank of Chinese Academy of Sciences, Shanghai, China), SMMC-7721 (Cell bank of Chinese Academy of Sciences, Shanghai, China), Huh7 (Cobioer, Nanjing, China), HepG2 (Cobioer, Nanjing, China), SK-Hep1 (Cell bank of Chinese Academy of Sciences, Shanghai, China), Bel-7404 (Cell bank of Chinese Academy of Sciences, Shanghai, China) and hepatocyte lines THLE-3 (Biovector, NTCC, Beijing, China) and HL-7702 (Cell bank of Chinese Academy of Sciences, Shanghai, China) were cultured in DMEM. Cells were treated with D-glucose or L-glucose (Sigma, St Louis, MO, USA) at a final concentration from 5.5 to 50 mM, PuGNAc (Sigma) at a final concentration of 25 μM, GlcNAc (Sigma) at a concentration of 4 mM, NaCl (Sangon, Shanghai, China) at a final concentration of 50 mM and Cycloheximide (CHX, sigma) at a final concentration of $50\,\mu g\,ml^{-1}$. The lentiviral-based OGT expression plasmid and OGT-shRNA (sh4) were purchased from Origene (Beijing, China), and the OGT-sh7 was purchased from Genechem (Shanghai, China) The expression plasmids encoding YAP-sh1&2, YAP, YAP-FLAG, TEAD4-Myc, CREB-HA, βTrCP-FLAG, c-Fos and the pUAS-Luc/TEAD-Gal4 system were constructed, as previously described by us[7,30,67]. The wild type, S208A, T213A, T216A, S217A, S227A, S229A and T241A-YAP-FLAG expression plasmids were constructed using pcDNA3.1(+) or pLJM-based lentiviral plasmid as the backbone. Promoter regions of human OGT, Nudt9 and SLC5A3 genes were PCR amplified from gDNA of Bel-7402 cells and cloned into pGL4.21 (Promega, Madison, WI, USA) vectors. The primers used for this study are listed in Supplementary Data 1.

**Immunohistochemistry and immunofluorescence (IF).** For immunohistochemistry (IHC), human liver cancer tissue microarray slides were purchased from U.S. Biomax (Rockville, MD, USA, #20810). The patient information provided by the TMA supplier is available in Supplementary Data 2. After deparaffinization and rehydration of the tissue sections, antigen retrieval was performed at 100 °C for 2 h in Tris-EDTA buffer, pH 6.0 (Beyotime). Endogenous peroxidases were blocked with 3% peroxide for 20 min and blocked in a buffer (5% BSA + 0.1% Triton X-100) and incubated overnight in primary antibodies. The primary antibodies used were anti-YAP (Abcam, Hong Kong, China, #ab52771, 1:100), anti-TEAD4 (Abcam, #97460, 1:100), anti-CTGF (Abcam, #6992, 1:500), anti-O-GlcNAc (Abcam, #ab2739, 1:100), anti-cleaved Caspase 3 (Cell Signaling Technology (CST), Boston, MA, USA, #9664, 1:500), anti-Ki67 (Abcam, #ab15580, 1:200) and the rabbit-origin anti-O-T241-YAP antibody (developed using O-GlcNAcylated peptide (GPLPDGWEQAMTQDG, O-GlcNAcylated at the Threonine site) as the antigen that was customized by Biolynx Biotechnology LTD (Hangzhou, China, 1:100)). The signals were detected with the Vectastain ABC kit (Vector Labs, Burlingame, CA, USA).

For IF, the cells were fixed with 4% paraformaldehyde for 15 min, washed with PBS and blocking buffer (3% FBS + 1% HISS + 0.1% Triton X-100), and then incubated overnight at 4 °C in primary antibodies. The primary antibodies used were anti-YAP (Abcam, #ab52771, 1:100 or Santa Cruz Biotechnology, Santa Cruz, CA, USA, #sc-101199, 1:100), anti-O-T241-YAP (Biolynx, 1:200), anti-O-GlcNAc (Abcam, #2739, 1:100), anti-OGT (Abcam, #ab184198, 1:100) and anti-p-YAP (Abcam, #ab76252, 1:200). Fluorescent Alexa-Fluor-488 or -555-conjugated secondary antibodies (Life technologies, Carlsbad, CA, USA) were used for detection.

**Western blotting.** For western blotting (WB), the proteins were resolved on SDS–polyacrylamide gel electrophoresis (SDS–PAGE) gels followed by standard WB. The primary antibodies used were anti-GAPDH (CST, #5174, 1:2,000), anti-OGT (Abcam, #ab184198, 1:1,000 or #ab177941, 1:1,000), anti-Myc-tag (CST, #2276, 1:1,000 or #2278, 1:1000), anti-YAP (Abcam, #ab52771, 1:1,000 or Santa Cruz, #sc-101199, 1:1,000), anti-p-YAP (Abcam, #ab76252, 1:2,000) anti-O-GlcNAc (Abcam, #ab2739, 1:1,000), anti-Ub (CST, #3936, 1:1,000), anti-CTGF (Abcam, #ab135812, 1:200), anti-HA (CST, #2367, 1:1,000 or #3724, 1:1,000), anti-FLAG (CST, #8146, 1:1,000 or #2368, 1:1,000), anti-βTrCP (CST, #4394, 1:1,000 or Santa Cruz, #sc-166492, 1:1,000), anti-SLC5A3 (Abcam, #ab113245, 1:1,000), anti-Nudt9 (Abcam, #ab170576, 1:500), anti-O-T241-YAP (Biolynx, 1:1,000), anti-c-Fos (CST, #2250, 1:1,000), anti-p-LATS1 (CST, #9157, 1:1,000) and anti-LATS1 (Abcam, #70561, 1:2,000). Blots were incubated with the appropriate secondary antibody (IRDye 800CW Goat anti-Mouse IgG (LICOR, Lincoln, NE, USA,#926-32210), IRDye 800CW Goat anti-Rabbit IgG (LICOR, #926-32211), anti-Rabbit IgG, HRP-linked Antibody (CST, #7074) or anti-Mouse IgG, HRP-linked Antibody (CST, #7076)), and the signals were detected by the Odyssey two-colour infrared laser imaging system (LICOR) or HRP-based chemiluminescence analysis. Some representative images of original blots are presented in Supplementary Fig. 11. The software ImageJ (v. 1.47, Bethesda, MD, USA) was used for densitometry of the western blots.

**Luciferase reporter assay.** Luciferase reporter vectors were stably cotransfected with a Renila luciferase expression plasmid into Bel-7402 and SMMC-7721 cells. Luciferase activities were detected using Dual-luciferase reagent (Promega)[7,30].

**Measurements of cell proliferation and Caspase3/7 activity.** For the methyl thiazol tetrazolium (MTT)-based cell-proliferation assay, Bel-7402 and SMMC-7721 cells (3,000 cells per well) were seeded in a 96-well plate, treated with 5 mg ml$^{-1}$ MTT 5 days later, and lysed in DMSO after 4 h. Absorbance was measured at 595 nm. Caspase3/7 activity was determined using Caspase 3/7 Glo luciferase reagent (Promega)[7,30].

**Soft agar colony formation assay.** For soft-agar colony formation assay, Bel-7402 and SMMC-7721 cells (6,000 cells per well) were seeded a 6-well plate of 0.3% agarose in DMEM media containing 10% FBS. Colonies from 12 fields of view were counted 2 weeks later[7,30].

**Quantitative RT-PCR and analysis of metabolites.** Relative quantitative RT-PCR (qPCR) was carried out using SYBR premix Ex Taq (TaKaRa, Japan).The primers used for qPCR are listed in Supplementary Data 1. Metabolites were analysed using corresponding kits from Sigma. Glucose was analysed with a kit from Applygen Biotechnology LTD (Beijing, China). ATP was analysed with a kit from Beyotime Biotechnology LTD (Haimen, China).

**Co-immunoprecipitation.** Cell lysates were incubated with protein A/G-Sepharose (Novex, Oslo, Norway) and western/IP lysis buffer (Beyotime) with or without 0.1% SDS (used for denatured IP (specifically for O-GlcNAc) and conventional IP, respectively). Immunoprecipitates were washed five times and then subjected to immunoblotting analysis. The antibodies used for IP were anti-YAP (Abcam, #ab52771, 1:100), anti-Myc (CST, #2278, 1:100), anti-HA (CST, #3724, 1:100), anti-FLAG (CST, #2368, 1:100 and #8146, 1:100), anti-O-GlcNAc (Abcam, #ab2739, 1:100) and anti-βTrCP (CST, #4394, 1:200)[33,68].

**Protein ligation assay.** A PLA was performed to identify a direct interaction between OGT and YAP using the Duolink *in situ* red starter kit (mouse/rabbit) (Sigma, Uppsala, Sweden). Cells were seeded on glass cover slips in 24-well plates overnight. In the second day, cells were fixed with 4% paraformaldehyde for 15 min. Then, the cells were blocked with the blocking buffer that was supplied by the manufacturer for 1 h and then were incubated overnight at 4 °C in primary antibodies. The primary antibodies were anti-YAP (Abcam, #ab52771, 1:100) and anti-OGT (Abcam, #ab184198, 1:100). On the third day, the PLA probe solution (supplied by the manufacturer) was added to each cell sample for 1 h at 37 °C, and then the Ligase–Ligase solution (supplied by the manufacturer) was added to each sample and incubated for 30 min at 37 °C. Finally, the amplification-polymerase solution (supplied by the manufacturer) was added to each sample for 100 min at 37 °C and was subjected to microscopic analysis. If the proximity between the two PLA probes is < 40 nm, then bright fluorescent emissions can be detected[69].

***In vitro* O-GlcNAcylation of YAP.** Reaction mixtures containing 1 μM purified human YAP protein (Abnova, Taiwan, # H00000116-P01), 0.125 μM human OGT (Abnova, H00008473-P01) and 10 μM UDP-GlcNAc (Sigma) in a buffer of 50 mM Tris/HCl (pH 7.5) and 1 mM DTT were incubated at 37 °C for 90 min. Then, co-IP was performed with anti-YAP antibodies (Abcam, #ab52771, 1:100) and detected by WB using anti-YAP (Santa Cruz, #sc-101199, 1:1000) and anti-O-GlcNAc (Abcam, #ab2739, 1:1000) antibodies.

**Enzymatic labelling of O-GlcNAc sites.** First, immunoprecipitated WT or T241A YAP with protein A/G-Sepharose (Novex, Oslo, Norway) were added into reaction buffer (20 mM HEPES, pH 7.9, 50 mM NaCl, 1 μM PuGNAc, and 5 mM MnCl$_2$ with protease and phosphatase inhibitors). Next, 2 μl of Gal-T1Y289L (Invitrogen, Carlsbad, CA, USA) and 2 μl of 0.5 mM UDP-GalNAz (Invitrogen) were added into reaction buffer to a volume of 20 μl. The reaction was performed overnight at 4 °C. The beads were washed twice with reaction buffer to remove excess UDP-GalNAz. The samples were then reacted with biotin alkyne (Invitrogen) or tetramethyl-6-carboxyrhodamine (TAMRA) alkyne (Invitrogen) according to the manufacturer's instructions. Biotin or TAMRA-labelled samples were finally detected by WB using HRP-labelled Streptavidin (Beyotime, #A0303, 1:1000) or antibodies against TAMRA (Invitrogen, #A6397, 1:1000).

**Mass spectrometry analysis.** Protein and posttranslational modification identification by mass spectrometry were performed in the Instrumental Analysis Center of Shanghai Jiaotong University (Shanghai, China). To determine the identities of the proteins co-purified with YAP, the corresponding protein bands in the Coomassie Brilliant blue gel were excised and in-gel digested with trypsin. The tryptic peptide digests of the proteins were analysed using a Capillary Electrophoresis/Nano-liquid chromatography (Nano-LC) system coupled with a Quadrupole-Time-of-Flight Mass Spectrometer (Bruker Daltonics, Leipzig,

Germany). An internal MASCOT 2.4.1 server (Matrix Science, Boston, MA, USA; http://www.matrixscience.com/) using the Swiss-Prot database was applied to identify peptides and their posttranslational modifications. The posttranslational modifications search parameter used was an increase of $+203.079373$ Da on serine/threonine residues (dynamic O-GlcNAcylation). The peptide mass tolerance was set to 20 p.p.m., and the fragment mass tolerance was set to 0.05 Da. The maximum of missed cleavage sites for trypsin was set to 2. The detailed information of all the peptides in the mass spectrometry analysis was presented in Supplementary Data 3.

**Statistical analysis.** Tests used to examine the differences between groups include Student's *t*-test, one-way and two-way ANOVA and $\chi^2$ test. A $P < 0.05$ was considered statistically significant.

**Data availability.** Mass spectrometry data have been deposited in the ProteomeXchange Consortium via the PRIDE partner repository under accession code PXD005992 (ref. 70). The authors declare that all the data supporting the findings of this study are available within the article and its Supplementary Information files, or available from the authors on request.

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

## Acknowledgements

This work was supported by the National Natural Science Foundation of China (grant numbers 81371913, 81301689, 81572330 and 81672332); the Shanghai Municipal Science and Technology Commission (grant number 14YF1412300) and a China central colleges and universities basic research–specific cross discipline grant (grant number 1501219096). We sincerely appreciate the technical assistance of Mass spectrometry by Dr Jingli Hou (Shanghai Jiaotong University, Shanghai, China).

## Author contributions

X.Z. designed the study, researched and analysed data. Y.Q., Q.W. and Y.C. researched data. S.Z. and X.L. collected and analysed clinical samples. G.Z.,Y.Z. and Y.C. contributed to discussion.. Y.Y. and Q.P. researched and analysed data.. J.W. designed the study. F.S. designed the study and wrote the manuscript.

## Additional information

**Competing interests:** The authors declare no competing financial interests.

