## [Peer Review File · Nature Communications]

Reviewers' comments:

Reviewer #1 (Remarks to the Author):

The manuscript reports some important information about the O-GlcNAcylation of YAP and the impact of that modification on YAP activity. The manuscript contains some useful information, with many important controls. However, in other cases, controls are lacking or poorly documented. The evidence that YAP is modified is very good. The stability of the protein is also clearly influenced by O-GlcNAcylation and that is also probably mediated by the E3-ligase β TrCP. Glucose is known to increase the O-GlcNAc modification in cancer cells and so it is not unreasonable to invoke the HBP as a mediator of the effects. It seems likely that most, if not all, of the effects are due to the enhanced stability of YAP upon O-GlcNAcylation. The suggestion that YAP upregulates the HBP and OGT is intriguing but here the evidence is much weaker and correlative. All in all, a nice paper with some interesting findings.

In general,

The manuscript has MANY grammatical errors and some lapses in scholarship. See examples below:

In Introduction:

Numerous citation errors:

For example:

This might be the molecular basis underlying elevated protein O-GlcNAcylation in diabetes 18. This reference is to a mitochondrial O-GlcNAc paper that does not deal directly with diabetes. Proper references here are Proc Natl Acad Sci U S A. 2002 Aug 6; 99(16): 10695–10699 and Proc Natl Acad Sci U S A. 2002 Apr 16; 99(8):5313-8.

Results:

The transcriptional activity of TEAD is the indicator of YAP activity

This statement probably should read activity of TEAD is one of the indicators of YAP activity. For growth, yes, for total transcriptional output, probably not.

Figure 1: This correlation, while intriguing may be recapitulated by other transcription factors. Is there a control for these analyses?

Figure 1: Correlation b/w YAP and O-GlcNAc signal on tissue array seems intriguing. There are varying levels of both YAP and O-GlcNAc and they seem to suggest that higher levels of YAP/O-GlcNAc correlated in HCC vs control tissues. This is recapitulated by Western. This also is inversely correlated with the amount of p-YAP present in the samples. (1) Is there a non-cancerous set of cell lines on the tissue microarray so a baseline can be established (as done with the HL-7702 in the Western?) (2) While showing statistics, it is not clear in D whether the error bars are from technical or biological replicates.

Figure 2: In Figure 1D, the data suggest that p-YAP levels are not different between HCC and normal tissues (it is the level of YAP that changes along with global O-GlcNAcylation). This seems to correlate with 2B in which inhibition of OGA correlates with decreased p-YAP but increased YAP and expression of downstream CTGF. (1) PugNAc is not an "activator" of O-GlcNAcylation, it is an inhibitor of OGA. (2) Does GlcNAc alone induce changes in YAP expression? (3) Need to show that there is an increase in global O-GlcNAc levels with treatments. (4) please note there are symbol errors in the figure.

Figure 4: Suggest that KD of OGT has reduction in OGT activity and yields increase in apoptotic cells. But, these phenotypes were reversed when YAP was over expressed. The claims for this figure need additional experiments. (1) What does addition of YAP over expression alone look like for these samples? (2) Overexpression of another protein would also be important as a control here to confirm that these findings are specific to YAP over expression. (3) In A wherein YAP is over expressed, it does not appear that there is an increase in YAP between the Mock and o/e. Does it just return to baseline expression of YAP? Is there a change in p-YAP as well?

In summary, this is a nice piece of work that could have important implications if vital experiments can be performed to test some of the ideas contained therein.

Reviewer #2 (Remarks to the Author):

Minor comments:

1. Lines 32-33: YAP, CREB, and TEAD need to be fully spelled as they appear in the manuscript for the first time. Furthermore, lines 269 and 280 need to be revised to correct the syntax.

Major comments:

1. Line 59: The authors provide intriguing data about a new mechanism of tumorigenesis linked to HCC associated with greater YAP and O-GlcNAcylation expression in tissue samples. Over 200 human HCC tissues using TMA analysis were screened. However, a part from the name of the institution where the tissues were collected, no further information about the patients was provided in the manuscript. Because one of the key findings positively links YAP expression to glucose levels in the cell lines Bel-7402 and SMMC-7721 (Figure S4A-B), it would be important to know etiology and metabolic background, especially the glycaemic levels of a representative number of patients. This information would improve the clinical impact of the findings provided in vitro and in the mouse model.

2. Line 64: The authors choose a broad spectrum of HCC cell lines and compare them with one non-cancerous hepatocyte line. A more representative number of non-cancerous hepatocyte cells should be included.

3. Line 338: The authors conclude that YAP, via O-GlcNAcylation, is crucial in liver tumorigenesis, particularly in diabetes-associated liver tumorigenesis. This conclusion is based on the studies conducted in mice models, whose diabetic condition is achieved by streptozotocin treatment (stated in line 241-242). However, the authors do not provide data showing the levels of insulin in the mice after treatment, nor they show any histopathology concerning this part of the study. Therefore, this conclusion lacks supporting information.

Based on these comments, we recommend that this manuscript be revised extensively. In its current state, the manuscript is not suitable for publication in Nature Communications.

Reviewer #3 (Remarks to the Author):

The manuscript reports a novel modification of YAP proto-oncogene protein, namely O-GlcNAcylation (OGNac) on Threonine 241 that is located within the conserved region of the second WW domain of YAP 1-2 isoform. Importantly, YAP that is decorated with OGNac on Thre-241 has a higher stability and pro-proliferative activity.

The study is original, well designed and clearly presented. The quality of the data is good and the burgeoning interest in the Hippo-YAP pathway and the not well understood role of YAP for propensity in gene amplification and causality in HCC should make this report of interest to a wide readership.

The Reviewer and two of his/her Colleagues who are experts in sugar metabolism suggest in concert the following changes to improve the manuscript.

1. It is likely, even from the visual inspection of the 3D structure of YAP WW domain in complex with its PPxY-containing cognate ligand that Thre 241 when decorated with a large sugar should disrupt YAP interaction with its cognate ligands, in particular with LATS (but also with AMOT, PTPN14) and result in nuclear localization of YAP to drive transcription of genes that induce proliferation. To document the mechanism of the YAP-OGNac-Thre-241, it would be important to show that either isolated second WW domain of YAP with OGNac-Thre-241 in vitro or YAP-OGNac-Thre-241 in cello show reduced propensity for interaction with LATS - PPxY peptide and better yet with LATS protein.

2. Please discuss better a correlation between YAP expression and global expression of OGNac proteins in HCC, in particular, if the localization of YAP-Thre-OGNac-241 in HCC as determined by IHC, for example, is changed as expected.

3. If we are correct in our interpretation, in Fig. 6C, the data may indicate that there is still another O-GluNac site in YAP? Please elaborate further on this point.

4. The following references could be added if space allows.

4.1. YAP WW domain was recently shown to be modified by Tyr phosphorylation in breast cancer models. This modification changes the ability of YAP WW domain to form complexes. Please consider discussing this report.

Li YW, Guo J, Shen H, Li J, Yang N, Frangou C, Wilson KE, Zhang Y, Mussell AL, Sudol M, Farooq A, Qu J, Zhang J. (2016) Phosphorylation of Tyr188 in the WW domain of YAP1 plays an essential role in YAP1-induced cellular transformation. *Cell Cycle*. Jul 18:0. [Epub ahead of print]

4.2. Original cloning of YAP, identification of the WW domain and its cognate PPxY ligands plus the presence of various isoforms of YAP could be referenced using original publications. Sudol, M. (1994). *Oncogene* 9, 2145-2152; Bork, P., and Sudol, M. (1994) *Trends in Biochem. Sci.* 19, 531-533; Chen H.I., and Sudol, M. (1995) *Proc. Natl. Acad. Sci. USA.* 92, 7819-7823. Gaffney, C.J., et al.,(2012) *Gene* 509, 215-222.

4.3. Since TAZ, a YAP paralogue does not have a second WW domain in vertebrates (except fish) and also YAP 1-1 isoform does not have a second WW domain, one could discuss why YAP is such a prevalent oncogene for liver cancer, compared to TAZ. The cloning of TAZ was by Mike Yaffe and his team at MIT and the report could be referenced in the discussion of TAZ. Kanai et al., (2000) *EMBO, J.*, 19, page 6778.

Reviewer #4 (Remarks to the Author):

This is a solid paper that establishes a causal relationship between Yap O-GlcNAcylation, lack of Yap phosphorylation, Yap stabilization, and tumorigenesis in HCC.

1. Better data should be obtained from the human HCCs with documentation of Yap O-GlcNAcylation and relationship to some objective criteria such as TEAD transcription, proliferation etc.

2. The mechanism for the inverse correlation between Yap O-GlcNAcylation and Yap phosphorylation at two distant sites should be determined.

#. The role of hyperglycemia in HCC is probably overstated. There is a small but significant increase in HCC in type I diabetes which is modeled by treatment with streptozotocin. The real increase is in type 2 diabetes with hyperinsulinemia, fatty liver, insulin resistance, and usually obesity.

Response to Reviewer #1

Question #1.1

The manuscript has MANY grammatical errors and some lapses in scholarship.

See examples below:

In Introduction: Numerous citation errors:

For example:

This might be the molecular basis underlying elevated protein

O-GlcNAcylation in diabetes ¹⁸. This reference is to a mitochondrial O-GlcNAc

paper that does not deal directly with diabetes. Proper references here are

Proc Natl Acad Sci U S A. 2002 Aug 6; 99(16): 10695–10699 and Proc Natl

Acad Sci U S A. 2002 Apr 16;99(8):5313-8.

Answer to Question #1.1

Firstly, we have asked the Springer Nature language editing service for proofreading our manuscript.

We have also carefully checked the citations throughout the manuscript, especially in the INTRODUCTION section. The following errors regarding citations have been found and corrected:

1. O-GlcNAcylation is a specific type of posttranslational modification catalyzed by O-GlcNAc transferase (OGT) ¹. The citation has been replaced by “Kreppel LK, Blomberg MA, Hart GW. Dynamic glycosylation of nuclear and cytosolic proteins. Cloning and characterization of a unique O-GlcNAc transferase with multiple tetratricopeptide repeats. J Biol Chem. 1997; 272: 9308-15.” (new ref. 1)
2. This might be the molecular basis underlying elevated protein O-GlcNAcylation in diabetes ¹⁸. The citation has been replaced by “McClain DA, Lubas WA, Cooksey RC, Hazel M, Parker GJ, Love DC, Hanover JA. Altered glycan-dependent signaling induces insulin resistance and hyperleptinemia. Proc Natl Acad Sci U S A. 2002; 99: 10695-9.” and “Vosseller K, Wells L, Lane MD, Hart GW. Elevated nucleocytoplasmic glycosylation by O-GlcNAc results in insulin resistance associated with defects in Akt activation in 3T3-L1 adipocytes. Proc Natl Acad Sci U S A. 2002; 99: 5313-8.” (new Ref. 18 and 19)

Question #1.2

Results:

The transcriptional activity of TEAD is the indicator of YAP activity. This statement probably should read activity of TEAD is one of the indicators of YAP activity. For growth, yes, for total transcriptional output, probably not.

Answer to Question #1.2

We agree with this suggestion from this reviewer, and have changed our statement more accurate and reasonable. The new statement is:

“Transcriptional activity of TEAD is one of the most important indicators of YAP activity.” (Page 5)

TEAD is required for YAP-induced cell growth, oncogenic transformation, and epithelial–mesenchymal transition [Zhao, et al., 2008]. Disruption of the TEAD–YAP complex suppresses the oncogenic activity of YAP [Liu, et al., 2012]. Hence, we believe testing TEAD transcription activity is a good way to indirectly reflect the activity of YAP.

References

Liu-Chittenden Y, Huang B, Shim JS, Chen Q, Lee SJ, Anders RA, Liu JO, Pan D. Genetic and pharmacological disruption of the TEAD-YAP complex suppresses the oncogenic activity of YAP. *Genes Dev.* 2012; 26: 1300-5. doi: 10.1101/gad.192856.112.

Zhao B, Ye X, Yu J, Li L, Li W, Li S, Yu J, Lin JD, Wang CY, Chinnaiyan AM, Lai ZC, Guan KL. TEAD mediates YAP-dependent gene induction and growth control. *Genes Dev.* 2008; 22: 1962-71. doi: 10.1101/gad.1664408.

Question #2

Figure 1: This correlation, while intriguing may be recapitulated by other transcription factors. Is there a control for these analyses?

Answer to Question #2

We chose transcription factor TEAD as the control. The reasons are listed below:

- 1) TEAD is one of the most important YAP-dependent transcription factors, and its transcription activity relies on YAP;

- 2) The aim of the present study is to investigate the correlation between O-GlcNAcylation and YAP and its contribution to liver tumorigenesis. Exclude the possibility that TEAD is also can be affected by O-GlcNAcylation is necessary to support the conclusion that the effects of O-GlcNAcylation on liver tumorigenesis are mainly via a YAP-dependent manner, but not via its YAP-dependent transcription factors.

The data in new Figure. 1a and Supplementary Fig. S1a demonstrate that the levels of O-GlcNAcylation correlate with the levels of YAP but not significantly correlate with the levels of TEAD. Moreover, data in new Figure. 1d-f also demonstrate that the TEAD expression has no significant differences between tumorous and adjacent normal liver tissues, and between established liver cancer cell lines and hepatocyte lines. These data further indicate that the effects of O-GlcNAcylation in liver cancer cells might rely on YAP but not its dependent transcription factors.

Question #3

Figure 1: Correlation b/w YAP and O-GlcNAc signal on tissue array seems intriguing. There are varying levels of both YAP and O-GlcNAc and they seem to suggest that higher levels of YAP/O-GlcNAc correlated in HCC vs control tissues. This is recapitulated by Western. This also is inversely correlated with the amount of p-YAP present in the samples. (1) Is there a non-cancerous set of cell lines on the tissue microarray so a baseline can be established (as done with the HL-7702 in the Western?) (2) While showing statistics, it is not clear in D whether the error bars are from technical or biological replicates.

Answer to Question #3

Because to the best of our knowledge, no tissue microarray regarding non-cancerous and cancerous liver cell lines is commercially available, we tested YAP and O-GlcNAc in a serial of cell lines including hepatocyte lines, THLE-3 and HL-7702, and liver cancer cell lines, SMMC-7721, Bel-7404, Bel-7402, HepG2, Huh7 and SK-Hep1, by IHC (new Figure. 1f, left panel). We found like the data from the Western blotting (new Figure. 1f, right two panels), data from IHC (new Figure. 1f, left panel) also showed that the positive correlation between YAP and O-GlcNAc in established cell lines. Notably, the

YAP/O-GlcNAc levels were much lower in hepatocyte lines than the levels in liver cancer cell lines; thereby further demonstrating that YAP/O-GlcNAc is elevated in cancerous cells.

The densitometry data from all the Western blots in the current study were collected from three independent experiments. Thereby the error bars in the Figure. 1d represent data from three biological replicates.

Question #4

Figure 2: In Figure 1D, the data suggest that p-YAP levels are not different between HCC and normal tissues (it is the level of YAP that changes along with global O-GlcNAcylation). This seems to correlate with 2B in which inhibition of OGA correlates with decreased p-YAP but increased YAP and expression of downstream CTGF. (1) PuGNac is not an "activator" of O-GlcNAcylation, it is an inhibitor of OGA. (2) Does GlcNAc alone induce changes in YAP expression? (3) Need to show that there is an increase in global O-GlcNAc levels with treatments. (4) please note there are symbol errors in the figure.

Answer to Question #4

Firstly, we mentioned in the manuscript that PuGNac is an inhibitor of OGA (Page 5).

In new Figures. 2a-b, the experiment using GlcNAc alone has been performed. We found treatment of GlcNAc alone can stimulate luciferase activity from pUAS-LUC/TEAD-Gal4 system (new Figure. 2a), YAP expression and global O-GlcNAcylation (new Figure. 2b), but simultaneously can reduce p-YAP levels (new Figure. 2b) in both Bel-7402 and SMMC-7721 cells. However, the stimulation by GlcNAc alone was not significant and also not as strong as the levels that stimulated by treatment of PuGNac alone or combined PuGNac and GlcNAc (new Figure. 2a-b); thereby in the follow-up experiments, GlcNAc alone was not used to stimulate O-GlcNAcylation in liver cancer cells.

In new Figure 2, global O-GlcNAcylation levels were examined, and the data are presented in 2b, 2d and 2e. As shown, treatment of PuGNac, combined PuGNac and GlcNAc and OGT overexpression, respectively, could induce a significant global O-GlcNAcylation. By contrast, OGT knockdown inhibited global O-GlcNAcylation in Bel-7402 and SMMC-7721 cells.

In the left panel of new Figure 2d, the “-” symbols have been aligned to a line. The symbol indicating statistical significance of Figure. 2e has been corrected.

Question #5

Figure 4: Suggest that KD of OGT has reduction in OGT activity and yields increase in apoptotic cells. But, these phenotypes were reversed when YAP was over expressed. The claims for this figure need additional experiments. (1) What does addition of YAP over expression alone look like for these samples? (2) Overexpression of another protein would also be important as a control here to confirm that these findings are specific to YAP over expression. (3) In A wherein YAP is over expressed, it does not appear that there is an increase in YAP between the Mock and o/e. Does it just return to baseline expression of YAP? Is there a change in p-YAP as well?

Answer to Question #5

In new Figure. 4a-d, addition of YAP over expression alone has been supplemented. YAP overexpression could increase cell proliferation and colony formation capacity, but inhibit apoptosis. However, its contributions to these phenotypes were not significant in liver cancer cells according to our statistics. These might due to YAP has already over-expressed or over-activated in liver cancer cells [Wang, et al., 2013; Wang, et al., 2015], the effects caused by addition of extra-YAP might not be obvious enough.

c-Fos is also an onco-protein that has been found to be overexpressed in liver cancer [Yuen, et al., 2001]. In new Supplementary Figure. S3, we noticed that knockdown of OGT had no significant effects on c-Fos (Figure. S3a). Further, knockdown of OGT could inhibit cell proliferation and colony formation capacity, whereas induce Caspase 3/7 activity in Bel-7402 and SMMC-7721

cells (Figure. S3b-d). However, the impaired transformative phenotypes could not be reversed by simultaneous overexpression of c-Fos (Figure. S3b-d). Thereby, we here confirm that the data from new Figure. 4 are specific to YAP over expression.

As shown in new Figure. 4a, overexpressing YAP alone could lead to a significant increase of YAP expression. However, due to the facts that knockdown of OGT reduces protein stability of YAP, simultaneous overexpression of YAP in Bel-7402 and SMMC-7721 cells with OGT knocked down was unable to reach to the levels in cells overexpressed with YAP alone. In order to reverse the YAP level to the baseline from “Mock”, we adjusted YAP levels by overexpressing YAP to the levels similar to the “Mock” as much as possible.

The p-YAP was also tested, and the data are shown in new Figure. 4a.

Reference

Wang, J. et al. Mutual interaction between YAP and CREB promotes tumorigenesis in liver cancer. *Hepatology* 58, 1011-1020, doi:10.1002/hep.26420 (2013).

Wang, J. et al. The membrane protein melanoma cell adhesion molecule (MCAM) is a novel tumor marker that stimulates tumorigenesis in hepatocellular carcinoma. *Oncogene* 34, 5781-5795, doi:10.1038/onc.2015.36 (2015).

Yuen, M. F., Wu, P. C., Lai, V. C., Lau, J. Y. & Lai, C. L. Expression of c-Myc, c-Fos, and c-jun in hepatocellular carcinoma. *Cancer* 91, 106-112. doi:10.1002/1097-0142(20010101)91:1<106::AID-CNCR14>3.0.CO;2-2 (2001).

Response to Reviewer #2

Question #1

Lines 32-33: YAP, CREB, and TEAD need to be fully spelled as they appear in the manuscript for the first time. Furthermore, lines 269 and 280 need to be revised to correct the syntax.

Answer to Question #1.1

YAP, CREB, and TEAD have been fully spelled as “Yes-associated protein” (Page 1), cAMP-response element binding protein” and “TEA domain transcription factor”, respectively when they firstly appear in the manuscript (Page 2).

The syntax in lines 269 and 280 has been corrected as follows:

Line 269: Next, we evaluated the expression of genes involved in the HBP.

Line 280: We found that YAP-induced mRNA expression and promoter activities of OGT, Nudt9 and SLC5A3 genes could be enhanced by simultaneous overexpression of either TEAD or CREB (Fig. 7c), further confirming that the HBP is transcriptionally regulated in a YAP-dependent manner.

Question #2

Line 59: The authors provide intriguing data about a new mechanism of tumorigenesis linked to HCC associated with greater YAP and O-GLCNAcylation expression in tissue samples. Over 200 human HCC tissues using TMA analysis were screened. However, a part from the name of the institution where the tissues were collected, no further information about the patients was provided in the manuscript. Because one of the key findings positively links YAP expression to glucose levels in the cell lines Bel-7402 and SMMC-7721 (Figure S4A-B), it would be important to know etiology and metabolic background, especially the glycaemic levels of a representative number of patients. This information would improve the clinical impact of the findings provided in vitro and in the mouse model.

Answer to Question #2

The liver cancer tissue microarray slides were purchased from U.S. Biomax (Rockville, MD, USA, #20810).

In the new Supplementary Table. S3, the limited patient information provided

by U.S. Biomax was summarized. Unfortunately, the glycaemic levels of each patient were not provided by U.S. Biomax.

In the new Supplementary Figure. S5o, we have evaluated p-YAP, YAP and global O-GlcNAcylation in 12 liver cancer patients. The patients #1-6 were liver cancer patients complicated with diabetes whereas the patients #7-12 were liver cancer patients without diabetes. The expressions of YAP and global O-GlcNAcylation were significantly higher in liver cancer patients complicated with diabetes compared to those without diabetes. By contrast, the p-YAP levels were much lower in liver cancer patients with diabetes compared to those without. These data suggest glycaemic levels are positively associated with the levels of YAP while negative associated with the p-YAP.

Question #3

Line 64: The authors choose a broad spectrum of HCC cell lines and compare them with one non-cancerous hepatocyte line. A more representative number of non-cancerous hepatocyte cells should be included.

Answer to Question #3

In addition to the hepatocyte line, HL-7702, another non-cancerous hepatocyte line, THLE-3 was added to compare the differences between hepatocyte and liver cancer cells. The data are shown in the new Figure. 1f and Supplementary Figure. S4p.

Question #4

Line 338: The authors conclude that YAP, via O-GlcNAcylation, is crucial in liver tumorigenesis, particularly in diabetes-associated liver tumorigenesis. This conclusion is based on the studies conducted in mice models, whose diabetic condition is achieved by streptozocin treatment (stated in line 241-242). However, the authors do not provide data showing the levels of insulin in the mice after treatment, nor they show any histopathology concerning this part of the study. Therefore, this conclusion lacks supporting information.

Answer to Question #4

The levels of serum-insulin along with serum-glucose from mice treated with either saline or streptozocin were examined, and the data are shown in the new Supplementary Figure. S5m. The histological changes of the liver and colon after treatment of either saline or streptozocin in mice are also shown in the new Supplementary Figure. S5n. It's found that treatment of streptozocin led to a more proliferative phenotype (have more dividing cells) in the liver compared to the saline-treated control (Supplementary Figure. S5n). However, no significant changes of phenotype between the treatments of saline and streptozocin were seen in the colon, suggesting the effects caused by streptozocin might be liver-specific.

Response to Reviewer #3

Question #1

It is likely, even from the visual inspection of the 3D structure of YAP WW domain in complex with its PPxY-containing cognate ligand that Thre 241 when decorated with a large sugar should disrupt YAP interaction with its cognate ligands, in particular with LATS (but also with AMOT, PTPN14) and result in nuclear localization of YAP to drive transcription of genes that induce proliferation. To document the mechanism of the YAP-OGNAc-Thre-241, it would be important to show that either isolated second WW domain of YAP with OGNAc-Thre-241 *in vitro* or YAP-OGNAc-Thre-241 *in cello* show reduced propensity for interaction with LATS - PPxY peptide and better yet with LATS protein.

Answer to Question #1

Thank you for your constructive suggestions. We have tried several times to synthesize the peptide of second WW domain of YAP, and this peptide contains potential O-GlcNAc site, Thr241. Unfortunately, it's very difficult to decorate this peptide at Thr241. Also, we are unable to synthesize the YAP protein with modification of O-GlcNAcylation at Thr241 *in vitro*, although we have tried a lot of times. For the above reasons, the only way to address this problem asked by this reviewer is to stimulate O-GlcNAcylation of YAP and see whether the YAP-LATS interaction is affected before and after mutation of Thr241 in liver cancer cells.

In new Figure 5d, stimulation of O-GlcNAcylation by combined treatment of PuGNAc and GlcNAc inhibited LATS1 binding to WT-YAP, however; this inhibitory effect was not significant when Thr241 was mutated. We also noticed that mutation of Thr241 in the YAP protein significantly increased YAP-LATS1 binding at basal level.

In new Figure 6d, the similar results were observed before and after treatments of high glucose.

Additionally, in new Supplementary Figure. S4o and S5e, we found the levels of p-LATS1 and total-LATS1 could not be affected by treatment of either combined PuGNAc and GlcNAc or high glucose. Thereby exclude the possibility that changes of YAP-LATS binding by O-GlcNAcylation and glucose might due to the changes of phosphorylation and expression of LATS.

Collectively, we propose that O-GlcNAcylation of YAP at Thr241 within the WW domain is accompanied by simultaneous reduction of the binding between YAP and LATS.

Question #2

Please discuss better a correlation between YAP expression and global expression of OGNAC proteins in HCC, in particular, if the localization of YAP-Thre-OGNAc-241 in HCC as determined by IHC, for example, is changed as expected.

Answer to Question #2

In DISCUSSION section, a better discussion on the correlation between YAP and global O-GlcNAcylation has been merged into the current version of manuscript, and is shown as follows in Page 20:

YAP stimulates tumorigenesis in liver cancer [Netsirisawan, et al. 2015; Sodi, et al., 2015]. O-GlcNAcylation is an important posttranslational modification of proteins, and plays pro-oncogenic roles in several types of cancer, including liver cancer [Ferrer, et al., 2014; Guo, et al., 2013; Huang, et al., 2013].

O-GlcNAcylation of onco-proteins Jun proto-oncogene (c-Jun) and Tribbles pseudokinase 2 (TRIB2) has been recently reported to stimulate liver tumorigenesis [Qiao et al., 2016; Yao et al., 2016]. Interestingly, both c-Jun and TRIB2 are nuclear proteins. In the current study, similar to c-Jun and TRIB2, we also found that both O-GlcNAcylated proteins and O-T241-YAP are primarily expressed in the nucleus. Therefore, we speculate that O-GlcNAcylation is more likely to occur in the nucleus, and this modification may easily enhance the expression and function of nuclear proteins, such as YAP. However, whether and how O-GlcNAcylation tends to occur in the nucleus is still unclear and needs to be further investigated.

In addition, we have developed an antibody that can specifically recognize O-GlcNAcylation of YAP at Thr241 (anti-O-T241-YAP). We used this antibody to evaluate O-GlcNAcylation of YAP in the same liver cancer microarray that has been used in new Figure. 1a. We found O-GlcNAcylation of YAP at Thr241 was significantly associated with the levels of both global O-GlcNAcylation and total-YAP (Supplementary Figure. S4q), further suggesting that global O-GlcNAcylation might reinforce YAP expression via O-GlcNAcylation of YAP at Thr241.

Moreover, we performed IF experiments in clinical liver cancer specimen, and found like the localization of YAP, O-GlcNAcylation occurred mainly in the nucleus, and higher expression of YAP correlated with higher global O-GlcNAcylation (new Supplementary Figure. S4r). Furthermore, O-GlcNAcylation of YAP at Thr241 was also found mainly in the nucleus (Supplementary Figure. S4r), where YAP exerts its pro-tumorigenic functions, and the data also suggested that nuclear portion of YAP can be O-GlcNAcylated at Thr241.

Reference

Ferrer, C. M. et al. O-GlcNAcylation regulates cancer metabolism and survival stress signaling via regulation of the HIF-1 pathway. *Molecular cell* 54, 820-831, doi:10.1016/j.molcel.2014.04.026 (2014).

Guo, K. et al. Translocation of HSP27 into liver cancer cell nucleus may be associated with phosphorylation and O-GlcNAc glycosylation. *Oncology reports* 28, 494-500, doi:10.3892/or.2012.1844 (2012).

Huang, X. et al. O-GlcNAcylation of cofilin promotes breast cancer cell invasion. *The Journal of biological chemistry* 288, 36418-36425, doi:10.1074/jbc.M113.495713 (2013).

Netsirisawan, P., Chokchaichamnankit, D., Srisomsap, C., Svasti, J. & Champattanachai, V. Proteomic Analysis Reveals Aberrant O-GlcNAcylation of Extracellular Proteins from Breast Cancer Cell Secretion. *Cancer genomics & proteomics* 12, 201-209 (2015).

Qiao, Y. et al. High Glucose Stimulates Tumorigenesis in Hepatocellular Carcinoma Cells Through AGER-Dependent O-GlcNAcylation of c-Jun. *Diabetes* 65, 619-632, doi:10.2337/db15-1057 (2016).

Sodi, V. L. et al. mTOR/MYC Axis Regulates O-GlcNAc Transferase Expression and O-GlcNAcylation in Breast Cancer. *Molecular cancer research : MCR* 13, 923-933, doi:10.1158/1541-7786.MCR-14-0536 (2015).

Yao, B. et al. Reciprocal regulation between O-GlcNAcylation and tribbles pseudokinase 2 (TRIB2) maintains transformative phenotypes in liver cancer cells. *Cellular signalling* 28, 1703-1712, doi:10.1016/j.cellsig.2016.08.003 (2016).

Question #3

If we are correct in our interpretation, in Fig. 6C, the data may indicate that there is still another O-GluNAc site in YAP? Please elaborate further on this point.

Answer to Question #3

Firstly, we agree with your opinion. We have discussed this point in the DISCUSSION section (Page 23). In new Figure. 5b and 6c, the results suggested that Thr241 might not be the only O-GlcNAc site within YAP1. It is common that several O-GlcNAc-modification sites are present in one glycoprotein. We suggest that Thr241 is the most pivotal O-GlcNAc site within YAP. In the MS experiment, the instrument has a certain sensitivity. The expression of O-T241-YAP reached a level that was detectable by MS.

However, the expression of other O-GlcNAc-modifications on YAP might not reach the cut-off value of the MS instrument. Moreover, when YAP was mutated on Thr241, the O-GlcNAc-modified level of YAP was significantly decreased. Furthermore, the T241A mutant YAP exhibited reduced protein stability and an impaired capacity to maintain transformative phenotypes compared to WT YAP. Therefore, we believe that although there might be other O-GlcNAc-modified sites within the YAP protein, Thr241 is the most important one.

Question #4.1

The following references could be added if space allows.

YAP WW domain was recently shown to be modified by Tyr phosphorylation in breast cancer models. This modification changes the ability of YAP WW domain to form complexes. Please consider discussing this report.

Answer to Question 4.1

The following context has been added into the DISCUSSION section (in Page 22):

Interestingly, Li et al. [Li, et al., 2016] reported that phosphorylation of tyrosine188 (Y188) in the YAP1-2 isoform stimulates YAP1-induced cellular transformation. Mutation of Y188 [especially replacement of Y to phenylalanine (F)] leads to a higher affinity of YAP for binding to its upstream negative regulators for cytoplasmic retention 51. Like Thr241, the Y188 site is also located in the conserved aromatic core of the second WW domain of YAP1. These findings further demonstrate that posttranslational modifications of WW domains may play significant roles in the function of YAP, specifically inducing changes in the ability of YAP to form complexes with other proteins. Whether Y188 phosphorylation and Thr241 O-GlcNAcylation have an impact on each other needs to be further investigated.

Reference

Li YW, Guo J, Shen H, Li J, Yang N, Frangou C, Wilson KE, Zhang Y, Mussell AL, Sudol M, Farooq A, Qu J, Zhang J. (2016) Phosphorylation of Tyr188 in

the WW domain of YAP1 plays an essential role in YAP1-induced cellular transformation. Cell Cycle. Jul 18:0. [Epub ahead of print]

Question #4.2

Original cloning of YAP, identification of the WW domain and its cognate PPxY ligands plus the presence of various isoforms of YAP could be referenced using original publications. Sudol, M. (1994). Oncogene 9, 2145-2152; Bork, P., and Sudol, M. (1994) Trends in Biochem. Sci. 19, 531-533; Chen H.I., and Sudol, M. (1995) Proc. Natl. Acad. Sci. USA. 92, 7819-7823. Gaffney, C.J., et al., (2012) Gene 509, 215-222.

Answer to Question 4.2

These citations have been added where the WW domain and PPxY ligands are firstly described in the manuscript (Page 21).

Question #4.3

Since TAZ, a YAP paralogue does not have a second WW domain in vertebrates (except fish) and also YAP 1-1 isoform does not have a second WW domain, one could discuss why YAP is such a prevalent oncogene for liver cancer, compared to TAZ. The cloning of TAZ was by Mike Yaffe and his team at MIT and the report could be referenced in the discussion of TAZ. Kanai et al., (2000) EMBO, J., 19, page 6778.

Answer to Question #4.3

The following context has been added into the DISCUSSION section (in Page 22-23):

The WW domain containing transcription regulator 1 (TAZ), a paralogue of YAP, has a structure similar to that of YAP. Unlike YAP, all the isoforms of TAZ in human cells do not have the second WW domain [Kanai, et al., 2000]. Comparatively speaking, YAP is a more prevalent oncoprotein in liver cancer. However, the function of TAZ in liver cancer is limited. We speculate that O-GlcNAcylation on the second WW domain of YAP plays an important role in promoting liver tumorigenesis, and this finding also supports the notion that YAP is more important than TAZ, because TAZ has only one WW domain that might not be O-GlcNAcylated. A series of studies [Komuro, et al., 2003; Oka, et

al., 2012] have also demonstrated that both YAP1-1 and YAP1-2 are present in liver tissue, and YAP1-2 has stronger transactivation activity compared to that of YAP1-1 [Komuro, et al., 2003]. Therefore, it is also not difficult to conclude that O-GlcNAcylation of the YAP1-2 proportion at its second WW domain might enhance the YAP pro-tumorigenic function contributed by both YAP1-1 and YAP1-2.

Reference

Kanai, F. et al. TAZ: a novel transcriptional co-activator regulated by interactions with 14-3-3 and PDZ domain proteins. *The EMBO journal* 19, 6778-6791, doi:10.1093/emboj/19.24.6778 (2000).

Komuro, A., Nagai, M., Navin, N. E. & Sudol, M. WW domain-containing protein YAP associates with ErbB-4 and acts as a co-transcriptional activator for the carboxyl-terminal fragment of ErbB-4 that translocates to the nucleus. *The Journal of biological chemistry* 278, 33334-33341, doi:10.1074/jbc.M305597200 (2003).

Oka, T., Schmitt, A. P. & Sudol, M. Opposing roles of angiotensin-like-1 and zona occludens-2 on pro-apoptotic function of YAP. *Oncogene* 31, 128-134, doi:10.1038/onc.2011.216 (2012).

Response to Reviewer #4

Question #1

Better data should be obtained from the human HCCs with documentation of Yap O-GlcNAcylation and relationship to some objective criteria such as TEAD transcription, proliferation etc.

Answer to Question #1

Because TEAD transcription is usually tested by a pUAS-LUC/TEAD-Gal4 system [Wang, et al., 2013; Wang, et al., 2013; Tang, et al., 2015], which is cell-based, thereby it's difficult to test TEAD transcription activity directly in tissues. CTGF is a well-established TEAD controlled gene [Zhao, et al., 2008], and its expression can indirectly reflect the transcription activity of TEAD. Due to the above reasons, we evaluated CTGF expression by IHC in the same liver

cancer samples whose levels of YAP and O-GlcNAc have already been tested (new Figure. 1a). The Ki67, a well-known proliferation marker, is also tested in the same samples (new Figure. 1a). The data indicate significant correlations between YAP and CTGF, between YAP and Ki67, between O-GlcNAc and CTGF, and between O-GlcNAc and Ki67 (new Figure. 1a and Supplementary Figure. S1a). However, TEAD expression was not correlated with either YAP or O-GlcNAc (Supplementary Fig. S1a). These data suggest that the YAP/O-GlcNAc correlation might be associated with cell proliferation, and CTGF expression might be associated with YAP/TEAD-dependent transcription activity but not directly correlated with TEAD expression. Furthermore, data from TMA stained by anti-O-T241-YAP antibodies also demonstrate a positive correlation between O-T241-YAP and CTGF, and between O-T241-YAP and Ki67 (Supplementary Figure. S4q), suggesting O-GlcNAcylation of YAP at Thr241 might be involved in the regulation of TEAD-dependent transcription and cell proliferation in liver cancer cells.

Reference

Tang, X. et al. CD166 positively regulates MCAM via inhibition to ubiquitin E3 ligases Smurf1 and betaTrCP through PI3K/AKT and c-Raf/MEK/ERK signaling in Bel-7402 hepatocellular carcinoma cells. *Cellular signalling* 27, 1694-1702, doi:10.1016/j.cellsig.2015.05.006 (2015).

Wang, J. et al. Mutual interaction between YAP and CREB promotes tumorigenesis in liver cancer. *Hepatology* 58, 1011-1020, doi:10.1002/hep.26420 (2013).

Wang, J. et al. TRIB2 acts downstream of Wnt/TCF in liver cancer cells to regulate YAP and C/EBPalpha function. *Molecular cell* 51, 211-225, doi:10.1016/j.molcel.2013.05.013 (2013).

Zhao, B. et al. TEAD mediates YAP-dependent gene induction and growth control. *Genes & development* 22, 1962-1971, doi:10.1101/gad.1664408 (2008).

Question #2

The mechanism for the inverse correlation between Yap O-GlcNAcylation and Yap phosphorylation at two distant sites should be determined.

Answer to Question #2

As shown in the new Figure 5d, stimulation of O-GlcNAcylation by treatment of PuGNAc and GlcNAc inhibited LATS1, the key kinase that can phosphorylate YAP at Ser127, binds with WT-YAP. However, this inhibitory effect was much reduced when the Thr241, the potential O-GlcNAc site, was mutated in both Bel-7402 and SMMC-7721 cells. We also noticed that mutation of Thr241 significantly increased YAP-LATS1 binding at basal level (new Figure. 5d). Moreover, we excluded that stimulation of O-GlcNAcylation influences phosphorylation of LATS1 (new Supplementary Fig. S4o), suggesting reduction of YAP-LATS1 binding might not due to alteration of LATS1 modification, instead, might due to the changes of YAP, especially modification at Thr241.

Interestingly, the Thr241 site is located within one WW domain in the YAP protein, and it's well known that the WW domain is essential for LATS1 binding with YAP [Hao, et al., 2008; Oka, et al., 2008]. These data suggest that O-GlcNAcylation of YAP at the Thr241 site within WW domain might prevent LATS1 interact with YAP for further phosphorylation.

Reference

Hao, Y., Chun, A., Cheung, K., Rashidi, B. & Yang, X. Tumor suppressor LATS1 is a negative regulator of oncogene YAP. *The Journal of biological chemistry* 283, 5496-5509, doi:10.1074/jbc.M709037200 (2008).

Oka, T., Mazack, V. & Sudol, M. Mst2 and Lats kinases regulate apoptotic function of Yes kinase-associated protein (YAP). *The Journal of biological chemistry* 283, 27534-27546, doi:10.1074/jbc.M804380200 (2008).

Question #3

The role of hyperglycemia in HCC is probably overstated. There is a small but significant increase in HCC in type I diabetes which is modeled by treatment

with streptazotycin. The real increase is in type 2 diabetes with hyperinsulinemia, fatty liver, insulin resistance, and usually obesity.

Answer to Question #3

Thank you for your suggestion. We have revised the manuscript according to your suggestion. But we still believe that hyperglycemia is one of the most important risk factors, which may cause high occurrence of liver cancer. In type 2 diabetes, hyperglycemia is still one of its major characteristics. Type 2 diabetes is associated with increased risk of liver cancer as numerous studies have reported [Chen et al., 2015; Qiao et al., 2014; Yang et al., 2013]. In one meta-analysis [Chen et al., 2015], authors have found that hepatitis C virus (HCV)-infected or cirrhotic patients with concomitant presence of type 2 diabetes are more likely to develop liver cancer than those without diabetes. Type 2 diabetes is also reported associated with poor prognosis of liver cancer [Qiao et al., 2014]. Thereby, as a common characteristic of both type 1 and type 2 diabetes, hyperglycemia might increase risk of the occurrence of liver cancer.

Reference

Chen, J., Han, Y., Xu, C., Xiao, T. & Wang, B. Effect of type 2 diabetes mellitus on the risk for hepatocellular carcinoma in chronic liver diseases: a meta-analysis of cohort studies. *European journal of cancer prevention : the official journal of the European Cancer Prevention Organisation* 24, 89-99, doi:10.1097/CEJ.0000000000000038 (2015).

Qiao, G., Le, Y., Li, J., Wang, L. & Shen, F. Glycogen synthase kinase-3beta is associated with the prognosis of hepatocellular carcinoma and may mediate the influence of type 2 diabetes mellitus on hepatocellular carcinoma. *PLoS one* 9, e105624, doi:10.1371/journal.pone.0105624 (2014).

Yang, W. S. et al. Prospective evaluation of type 2 diabetes mellitus on the risk of primary liver cancer in Chinese men and women. *Annals of oncology : official journal of the European Society for Medical Oncology / ESMO* 24, 1679-1685, doi:10.1093/annonc/mdt017 (2013).

REVIEWERS' COMMENTS:

Reviewer #1 (Remarks to the Author):

Most of the reviewer requests were addressed.

Reviewer #2 (Remarks to the Author):

The additional experiments to answer the reviewer's main concerns clarified the proposed mechanism in which Yap O-GlcNAcylation is implicated in liver tumorigenesis. Concern #2 referred to the authors' response to reviewer number 2: While the authors could not provide a complete metabolic panel for the HCC cases provided in Supplementary Table S3, they looked at a set of 12 patients with HCC with or without history of diabetes, and found that Yap and O-GlcNAcylation were higher in patients with a background of diabetes. To address the concern #3, regarding the number of cell lines used in the study, the authors included an additional non-cancerous cell line, THLE-3. Concern #4: The authors incorporate the serum-insulin and serum-glucose tests to assess the diabetic mice phenotype upon streptozotocin treatment.

Overall, the additional information provided by the authors is satisfactory therefore the manuscript is suitable for publication.

Reviewer #3 (Remarks to the Author):

The authors addressed all my comments and queries very well.

Reviewer 4 only commented for the editors and was supportive of publication.